# Ecdysone coordinates plastic growth with robust pattern in the developing wing

André Nogueira Alves[1,2†], Marisa Mateus Oliveira[1†], Takashi Koyama[1,3], Alexander Shingleton[4*‡], Christen Kerry Mirth[1,2*‡]

[1]Instituto Gulbenkian de Ciência, Oeiras, Portugal; [2]School of Biological Sciences, Monash University, Melbourne, Australia; [3]Department of Biology, University of Copenhagen, Copenhagen, Denmark; [4]Department of Biological Sciences, University of Illinois at Chicago, Chicago, United States

**Abstract** Animals develop in unpredictable, variable environments. In response to environmental change, some aspects of development adjust to generate plastic phenotypes. Other aspects of development, however, are buffered against environmental change to produce robust phenotypes. How organ development is coordinated to accommodate both plastic and robust developmental responses is poorly understood. Here, we demonstrate that the steroid hormone ecdysone coordinates both plasticity of organ size and robustness of organ pattern in the developing wings of the fruit fly *Drosophila melanogaster*. Using fed and starved larvae that lack prothoracic glands, which synthesize ecdysone, we show that nutrition regulates growth both via ecdysone and via an ecdysone-independent mechanism, while nutrition regulates patterning only via ecdysone. We then demonstrate that growth shows a graded response to ecdysone concentration, while patterning shows a threshold response. Collectively, these data support a model where nutritionally regulated ecdysone fluctuations confer plasticity by regulating disc growth in response to basal ecdysone levels and confer robustness by initiating patterning only once ecdysone peaks exceed a threshold concentration. This could represent a generalizable mechanism through which hormones coordinate plastic growth with robust patterning in the face of environmental change.

## Editor's evaluation

This article is a carefully done assessment of the role of the moulting hormone ecdysone in coordinating growth and patterning of the wing imaginal disc in the final larval instar of *Drosophila* with nutritional input. Importantly, the authors find that growth is only partially dependent on the ecdysteroid titre, whereas the onset of bristle patterning is dependent on a threshold level that is different for different genes.

## Introduction

Developing animals respond to changes in their environment in a multitude of ways, for example, altering how long and how fast they grow, the time it takes them to mature, and their reproductive output (*Nylin and Gotthard, 1998*; *West-Eberhard, 1989*). Other aspects of their phenotype, however, must be unresponsive to environmental change to ensure that they function correctly regardless of environmental conditions. This presents a particular problem for morphological traits of developing animals. For any given trait, some aspects, such as final organ size, vary with changes in the environment, a phenomenon termed plasticity (*Beldade et al., 2011*; *Koyama et al., 2013*; *Shingleton, 2010*; *Mirth and Shingleton, 2019*; *Nijhout et al., 2017*). Other aspects, like patterning the cell types within an organ necessary for it to function, remain constant across environmental conditions

*For correspondence:
ashingle@uic.edu (AS);
christen.mirth@monash.edu
(CKM)

[†]These authors contributed
equally to this work
[‡]These authors also contributed
equally to this work

**Competing interest:** The authors declare that no competing interests exist.

and are thus termed robust (*Mirth and Shingleton, 2019*; *Nijhout et al., 2017*; *Félix and Barkoulas, 2015*; *Félix and Wagner, 2008*). For many organs, growth and patterning occur at the same time during development, and may even be regulated by the same hormones (*Mirth and Shingleton, 2019*). How then do organs achieve plasticity in size while maintaining the robustness of pattern?

If we want to extract general principles of how organisms regulate their development in variable environments, we need to understand how developmental processes unfold over time. Several recent studies that have applied systems approaches to development offer excellent examples, frequently employing methods to quantify how gene expression patterns change over time. These studies have used the dynamic changes in expression patterns to uncover the rules governing how insects build their segments (*Surkova et al., 2009a*; *Surkova et al., 2009b*), how the gene regulatory network underlying segmentation evolves (*Clark, 2017*; *Clark and Akam, 2016*; *Clark and Peel, 2018*; *Crombach et al., 2016*; *Verd et al., 2018*; *Wotton et al., 2015*), how morphogen gradients scale across organs and bodies (*Zhou et al., 2012*; *Almuedo-Castillo et al., 2018*; *Zhu et al., 2020*; *Schwank et al., 2011*; *Wartlick et al., 2011*; *Hamaratoglu et al., 2011*), how sensory organs are positioned within epithelia (*Corson et al., 2017*), and how somites and digits form in vertebrates (*Raspopovic et al., 2014*; *Dubrulle et al., 2001*; *Baker et al., 2006*). The power of these approaches is that they provide a framework for understanding how genes interact within a network to generate a pattern that can be applied across a variety of contexts.

The success of these studies is, in part, due to the fact that the gene regulatory networks underlying each of these processes have been well described in their respective developmental contexts. In contrast, the gene regulatory networks governing growth and patterning at later stages of development, even at later stages of embryonic development, are not as well resolved. If we further complicate this by comparing development across environmental conditions and even across traits, approaches that rely on understanding the configuration of gene regulatory networks become much more difficult to implement.

Nevertheless, we can still use the principle of comparing the dynamics of developmental processes across environments to gain useful insights into the relationship between plasticity and robustness. Many types of environmental conditions impact organ development to induce changes in body and organ size. Malnutrition or starvation reduces growth rates in all animals, resulting in smaller body and organ sizes (*Nijhout, 2003*; *Nijhout et al., 2014*; *Mirth and Shingleton, 2012*). Similarly, changing temperature can alter animal growth. In insect species, rearing animals in warmer conditions results in smaller adult body sizes when compared to animals reared under cooler conditions (*Azevedo et al., 2002*; *David et al., 1994*; *French et al., 1998*; *James et al., 1997*; *Partridge et al., 1994*; *Grunert et al., 2015*; *Reynolds and Nottingham, 1985*; *Thomas, 1993*). Other factors like oxygen availability and the presence of toxic or noxious compounds also act to alter animal sizes (*Callier and Nijhout, 2011*; *Callier et al., 2013*; *Glendinning, 2003*). Examining how organ growth and patterning progress across these environmental conditions helps us to understand how these two processes are coordinated.

We already have some understanding of the mechanisms that regulate growth and patterning in response to changing environmental conditions. The genetic mechanisms underlying plasticity in growth are best elucidated in insects. In insects, changes in available nutrition affect the synthesis and secretion of the conserved insulin-like peptides (*Wu and Brown, 2006*; *Brogiolo et al., 2001*; *Ikeya et al., 2002*). Insulin-like peptides bind to the insulin receptor in target tissues and activate the insulin signalling cascade, ultimately leading to increased growth (*Brogiolo et al., 2001*; *Chen et al., 1996*; *Yenush et al., 1996*). Starvation reduces the concentration of insulin-like peptides in the hemolymph, or insect blood, and the resulting decrease in insulin signalling causes organs to grow more slowly (*Ikeya et al., 2002*; *Géminard et al., 2009*).

While changes in insulin signalling are known to affect organ size, they have little effect on organ pattern (*Weinkove and Leevers, 2000*). However, studies in the fruit fly *Drosophila melanogaster* have shown that, at least in this insect, insulin acts to control the synthesis of a second developmental hormone, the steroid hormone ecdysone (*Caldwell et al., 2005*; *Colombani et al., 2012*; *Mirth et al., 2005*; *Koyama et al., 2014*). Most of the body and organ growth in *D. melanogaster* occurs in the third, and final, larval instar, after which the animal initiates metamorphosis at pupariation. Either starving or reducing insulin signalling early in the third instar delays the timing of ecdysone synthesis, thereby prolonging the length of the third instar and the time it takes to metamorphose (*Caldwell*

*et al., 2005*; *Colombani et al., 2012*; *Mirth et al., 2005*; *Koyama et al., 2014*; *Shingleton et al., 2005*).

In addition to its effects on developmental time, ecdysone controls the growth of the developing adult organs (*Stieper et al., 2008*; *Herboso et al., 2015*; *Gokhale et al., 2016*; *Dye et al., 2017*). In *D. melanogaster* larvae, many of the adult organs form and grow inside the larvae as pouches of cells called imaginal discs. If ecdysone synthesis is reduced or if the glands that produce ecdysone, the prothoracic glands (PG), are ablated, these imaginal discs grow at greatly reduced rates (*Herboso et al., 2015*; *Mirth et al., 2009*).

Ecdysone signalling also regulates organ patterning. Reducing ecdysone signalling in either the wing imaginal disc or the developing ovary causes substantial delays in their patterning (*Herboso et al., 2015*; *Mirth et al., 2009*; *Mendes and Mirth, 2016*; *Gancz et al., 2011*). In the wing disc, reducing ecdysone signalling stalls the progression of patterning of sensory bristles (*Herboso et al., 2015*; *Mirth et al., 2009*). Similarly, in the ovary terminal filament cell specification and the rate of terminal filament addition both require ecdysone to progress normally (*Mendes and Mirth, 2016*; *Gancz et al., 2011*). Given its role in both the patterning and the growth of imaginal discs and ovaries, ecdysone is potentially a key coordinator of plastic growth and robust pattern.

Characterizing organ growth rates is experimentally straightforward, requiring only accurate measurement of changes in organ size over time. To quantify the progression of organ patterning, however, requires developing a staging scheme. We previously developed such a scheme for the wing imaginal disc in *D. melanogaster*. This scheme makes use of the dynamic changes in expression from the moult to the third instar to pupariation of up to seven patterning-gene products in the developing wing (*Oliveira et al., 2014*). Two of these patterning-gene products, Achaete and Senseless, can be classed into seven different stages throughout third-instar development (*Oliveira et al., 2014*, *Figure 1A*), providing us with the ability to quantify the progression of wing disc pattern over a variety of conditions. In short, by describing patterning on a near-continuous scale, our scheme not only allows us to determine under what conditions patterning is initiated, but also the rate at which it progresses.

The ability to simultaneously quantify both organ growth and pattern allows us to generate, and test, hypotheses regarding how ecdysone coordinates plastic growth with robust patterning. One hypothesis is that growth and patterning occur at different times, with ecdysone driving growth first then pattern later, or vice versa (*Mirth and Shingleton, 2019*). If this were true, we would expect to identify an interval where ecdysone concentrations primarily affected growth and a second interval where they affected mostly pattern (*Figure 1B*). There is some precedence for this idea; most of the patterning in the wing discs and ovaries of *D. melanogaster* occurs 15 hr after the moult to the third larval instar (*Mendes and Mirth, 2016*). Similarly, wing discs are known to grow faster in the early part of the third instar and slow their growth in the mid-to-late third instar (*Shingleton et al., 2008*). As a second hypothesis, ecdysone could coordinate plastic growth with robust pattern if the impacts of ecdysone on one of these processes depended on its effects on the other. For example, morphogens are known to regulate both growth and patterning of the wing. If ecdysone controlled the action of morphogens, we would expect the progression of patterning to be tightly coupled to growth over time, with different aspects of patterning being initiated at different disc sizes (*Figure 1C*). Finally, a third hypothesis is that ecdysone regulates the growth and patterning of the wing discs independently, and that each process responds in a qualitatively and quantitatively different manner to ecdysone (*Mirth and Shingleton, 2019*). As an example of this, we might see that growth rates increase in a graded response to increasing ecdysone while patterning shows threshold responses, or vice versa. If this were the case, we would expect that growth and the progression of pattern would be uncoupled over time (*Figure 1D*).

Here, we test these hypotheses of whether and how ecdysone co-regulates plastic growth and robust pattern in wing imaginal discs in *D. melanogaster*. We blocked the production of ecdysone by genetically ablating the PG (*Herboso et al., 2015*) and quantified the effects on growth and patterning rates throughout the third instar. We then manipulated the rate of ecdysone synthesis by up- or down-regulating the activity of the insulin-signalling pathway in the PG (*Mirth et al., 2005*; *Koyama et al., 2014*) to test how this alters the relationship between disc size and disc pattern. Finally, we tested our hypotheses about how a single steroid can regulate both plastic growth and robust patterning by conducting dose-response experiments under two nutritional conditions. These studies provide a

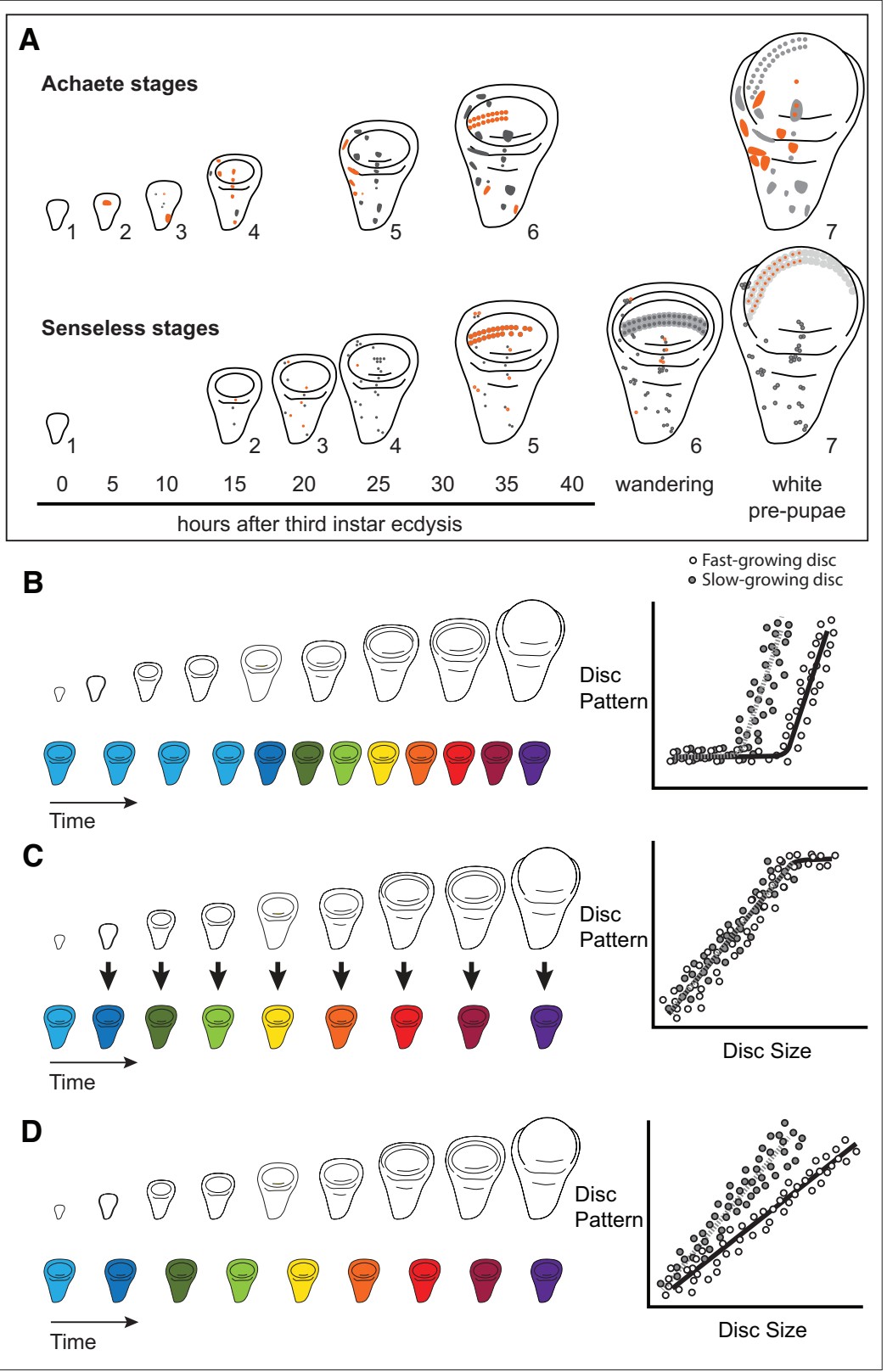

**Figure 1.** Quantitative assessments of the progression of patterning allow us to test hypotheses about the relationship between the size and patterning stage of the developing wing. (**A**) The staging scheme developed by *Oliveira et al., 2014* to quantify the progression of Achaete and Senseless pattern. The pattern elements shown in orange are diagnostic for each stage, which is indicated by the number beside the disc. (**B–D**) The relationship

*Figure 1 continued on next page*

*Figure 1 continued*

between wing disc size and patterning stage (represented as wing discs progressing through a series of colours) if (**B**) Hypothesis 1: wing discs grow first and then initiate pattern; (**C**) Hypothesis 2: wing disc patterning is regulated by wing disc size (arrows); and (**D**) Hypothesis 3: wing disc pattern and growth are regulated at least partially independently.

foundation for a broader understanding of how developmental hormones coordinate both plastic and robust responses across varying environmental conditions during animal development.

## Results

### Ecdysone is necessary for the progression of growth and patterning

To understand how ecdysone affects the dynamics of growth and patterning, we needed to be able to precisely manipulate ecdysone concentrations. For this reason, we made use of a technique we developed previously to genetically ablate the PGs (referred to as PGX) (**Herboso et al., 2015**). This technique pairs the temperature-sensitive repressor of GAL4, GAL80$^{ts}$, with a PG-specific GAL4 (*phm-GAL4*) to drive an apoptosis-inducing gene (*UAS-GRIM*). GAL80$^{ts}$ is active at 17°C, where it represses GAL4 action, but inactive above 25°C, which allows *phm-GAL4* to drive expression of *UAS-GRIM* and ablate the PG (**McGuire et al., 2003**; **McGuire et al., 2004**). Because ecdysone is required at every moult, we reared larvae from egg to the third larval instar (L3) at 17°C to repress GAL4, then shifted the larvae to 29°C at the moult to the third instar to generate PGX larvae.

PGX larvae had significantly reduced ecdysteroid titres than control genotypes (**Figure 2—figure supplement 1**). This method of reducing ecdysteroid concentration in the larvae allows us to examine how reducing ecdysone titres affects disc size and pattern in third-instar wing imaginal discs and manipulate ecdysone concentrations by adding it back in specific concentrations to the food (**Herboso et al., 2015**). For simplicity, all the data from the two control strains (either the *phm-GAL4; GAL80ts* or *UAS-GRIM* parental strain crossed to w$^{1118}$) were pooled in all analyses.

Insect wing discs show damped exponential, or fast-then-slow, growth dynamics (**Shingleton et al., 2008**; **Nijhout and Wheeler, 1996**). These types of growth dynamics have frequently been modelled using a Gompertz function, which assumes that exponential growth rates slow down with time. The growth of wing discs from control and PGX larvae shows the same pattern, with a Gompertz function providing a significantly better fit to the relationship between log disc size and time than a linear function (ANOVA, linear vs. Gompertz model, n > 93, *F* > 65, p<0.001 for discs from both PGX and control larvae). Growth of the discs, however, followed a significantly different trajectory in PGX versus control larvae (**Figure 2**, **Supplementary file 1a**). In control larvae, discs continue to grow until 42 hr after ecdysis to the third instar (AEL3) when the larvae pupariate. In contrast, the wing imaginal discs of the PGX larvae grow at slower rates between 0 and 25 hr AEL3 (**Figure 2**, **Supplementary file 1**a) and stop growing at approximately 25 hr AEL3 at a significantly smaller size. This is despite the fact that PGX larvae do not pupariate, and so disc growth is not truncated by metamorphosis.

We next explored how the loss of ecdysone affected the progression of wing patterning. We used the staging scheme that we previously devised in **Oliveira et al., 2014** to quantify the progression of wing disc patterning in PGX and control larvae. We selected two gene-products from this scheme, Achaete and Senseless, as they each progress through seven stages throughout the third instar. Further we can stain for both antigens in the same discs, which allowed us to compare disc size, Achaete stage, and Senseless stage in the same sample.

The progression of Achaete patterning was best fit by a Gompertz function for discs from both PGX and control larvae (ANOVA, linear versus Gompertz model, n > 48, *F* > 10.4, p=0.002) (**Oliveira et al., 2014**) and was significantly affected by reduced ecdysone titres in PGX larvae. In control larvae, the wing discs progressed to Achaete stage 6 or 7 out of seven stages by 42 hr AEL3, while in PGX larvae, discs of the same age had not passed Achaete stage 3, and had not matured past Achaete stage 5 by 92 hr AEL3 (**Figure 3A**, **Supplementary file 1**b). The progression of Senseless patterning was best fit by a linear model, but again was significantly affected by reduced ecdysone titres. In control larvae, most discs had progressed to Senseless stage 6 out of seven stages by 42 hr AEL3, while no disc progressed past Senseless stage 2 by 92 hr AEL3 (**Figure 3B**, **Supplementary file 1**c).

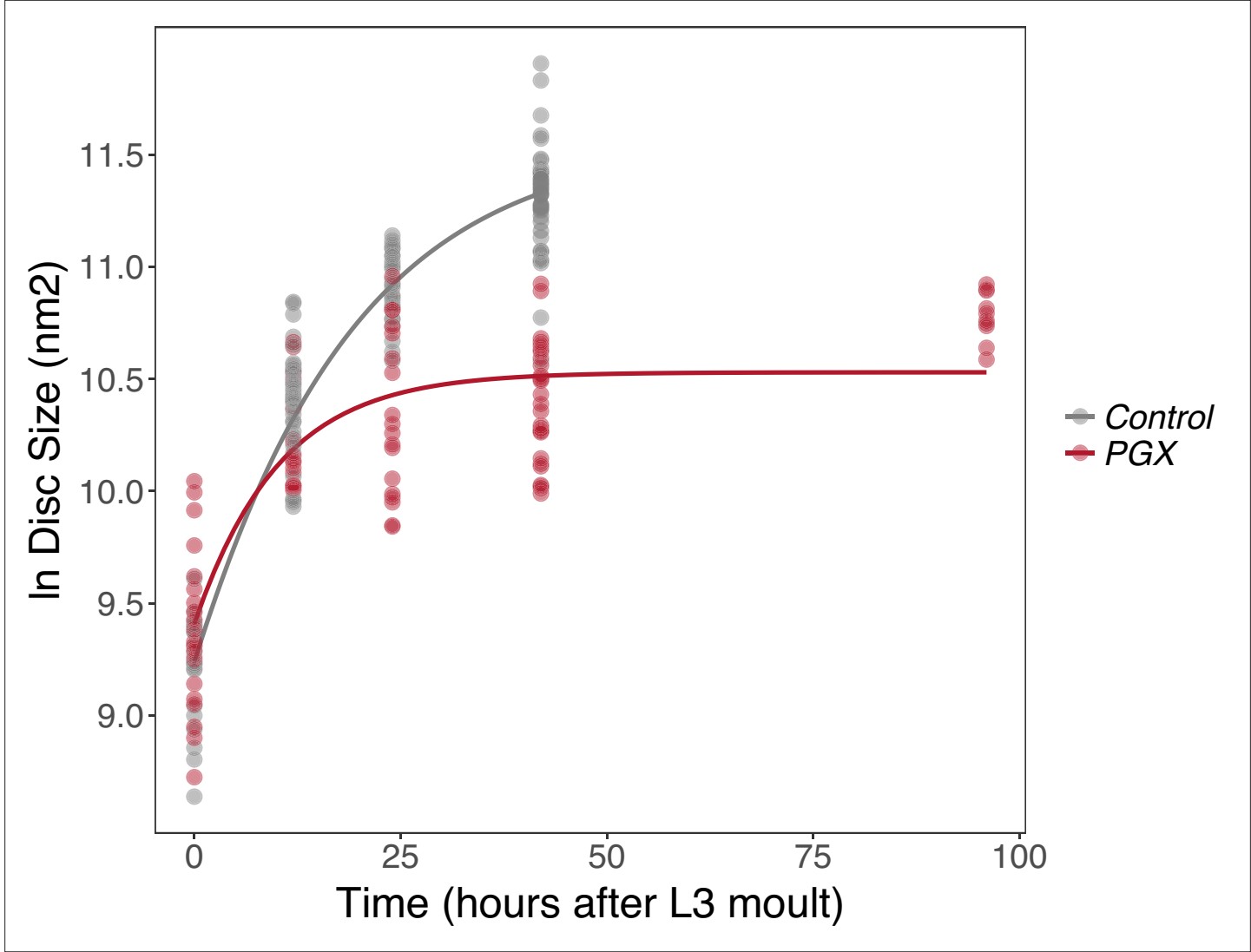

**Figure 2.** Growth rates of wing discs are reduced in larvae with genetically ablated prothoracic glands (PGX) versus control larvae. Curves are Gompertz functions of disc size against time (hours after the third larval instar (L3) moult). Parameters for the curves are significantly different between PGX and control (*Supplementary file 1*a). Control genotypes are the pooled results from both parental controls (either the *phm-GAL4; GAL80ts*, or *UAS-GRIM* parental strain crossed to w[1118]). Each point represents the size of an individual wing disc. $N_{PGX} = 95$, $N_{Control} = 125$ across all time points.

The online version of this article includes the following figure supplement(s) for figure 2:

**Figure supplement 1.** Ecdysteroid titres in genetically ablated prothoracic glands (PGX) and control larvae.

We found no evidence of temporal separation between wing disc growth and the progression of pattern (compare *Figures 2 and 3*). Both growth and patterning progressed at steady rates throughout most of the third instar in control larvae, slowing down only at the later stages of development. Thus, the hypothesis that ecdysone coordinates plastic growth with robust pattern by acting on each process at different times (*Figure 1B*; Hypothesis 1) is not correct.

To confirm that reduced ecdysone titres were responsible for delayed patterning, and not a systemic response to the death of the glands, we performed a second experiment where we added either the active form of ecdysone, 20-hydroxyecdysone (20E), or ethanol (the carrier) back to the food. PGX and control larvae were transferred onto either 20E or ethanol food and allowed to feed for 42 hr, after which we dissected their wing discs and examined their size and pattern. On the control (ethanol) food, wing discs from PGX larvae were smaller (*Figure 4A*, *Supplementary file 1*d) and showed reduced patterning for both Achaete (*Figure 4B*, *Supplementary file 1*d) and Senseless (*Figure 4C*, *Supplementary file 1*d) when compared to control genotypes. Adding 0.15 mg of 20E/mg food fully

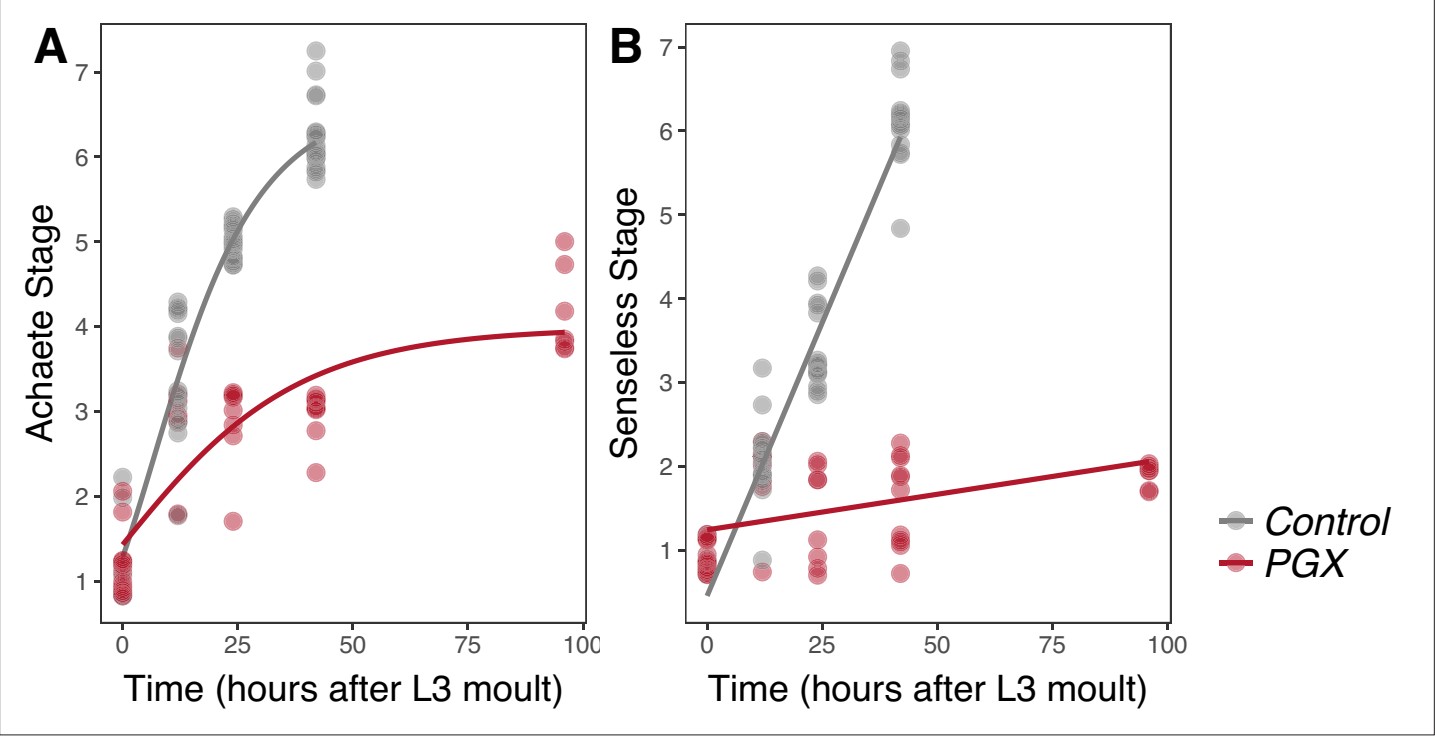

**Figure 3.** Achaete and Senseless patterning of wing discs is delayed in larvae with genetically ablated prothoracic glands (PGX) versus control larvae. (**A**) Curves are Gompertz functions of Achaete stage against time (hours after the third instar (L3) moult). Parameters for the curves are significantly different between PGX and control (*Supplementary file 1*b). (**B**) Lines are linear regression of Senseless stage against time (hours after the L3 moult). Parameters for the lines are significantly different between PGX and control (*Supplementary file 1*c). Control genotypes are the pooled results from both parental controls (either the *phm-GAL4; GAL80ts,* or *UAS-GRIM* parental strain crossed to $w^{1118}$). For Achaete: $N_{PGX}$ = 50, $N_{Control}$ = 61, for Senseless: $N_{PGX}$ = 52, $N_{Control}$ = 54 across all time points.

restored disc size, and Achaete and Senseless pattern, such that they were indistinguishable from control genotypes fed on 20E-treated food.

Collectively, these data indicate that ecdysone is necessary for the normal progression of growth and patterning in wing imaginal discs. The loss of ecdysone has a more potent effect on patterning, however, which is effectively shutdown in PGX larvae, than on disc growth, which continues, albeit at a slower rate, for the first 24 hr of the third instar in PGX larvae.

## Ecdysone rescues patterning and some growth in wing discs of yeast-starved larvae

The observation that ecdysone is necessary to drive both normal growth and patterning suggests that it may play a role in coordinating growth and patterning across environmental conditions. However, to do so it must lie downstream of the physiological mechanisms that sense and respond to environmental change. As discussed above, ecdysone synthesis is regulated by the activity of the insulin-signalling pathway, which is in turn regulated by nutrition. Starving larvae of yeast early in the third instar both suppress insulin signalling and inhibit growth and patterning of organs (*Mirth et al., 2009*; *Mendes and Mirth, 2016*). We explored whether ecdysone was able to rescue some of this inhibition by transferring larvae immediately after the moult to 1% sucrose food that contained either 20E or ethanol and comparing their growth and patterning after 24 hr to wing discs from larvae fed on normal food. Both the PGX and control genotype failed to grow and pattern on the 1% sucrose with ethanol (*Figure 5A–C*, *Supplementary file 1*e). Adding 20E to the 1% sucrose food rescued Achaete and Senseless patterning in both the control and the PGX larvae to levels seen in fed controls (*Figure 5B*, *Supplementary file 1*e). 20E also partially rescued disc growth in PGX larvae, although not to the levels of the fed controls (*Figure 5A*). Collectively, these data suggest that the effect of nutrition on growth and patterning is at least partially mediated through ecdysone.

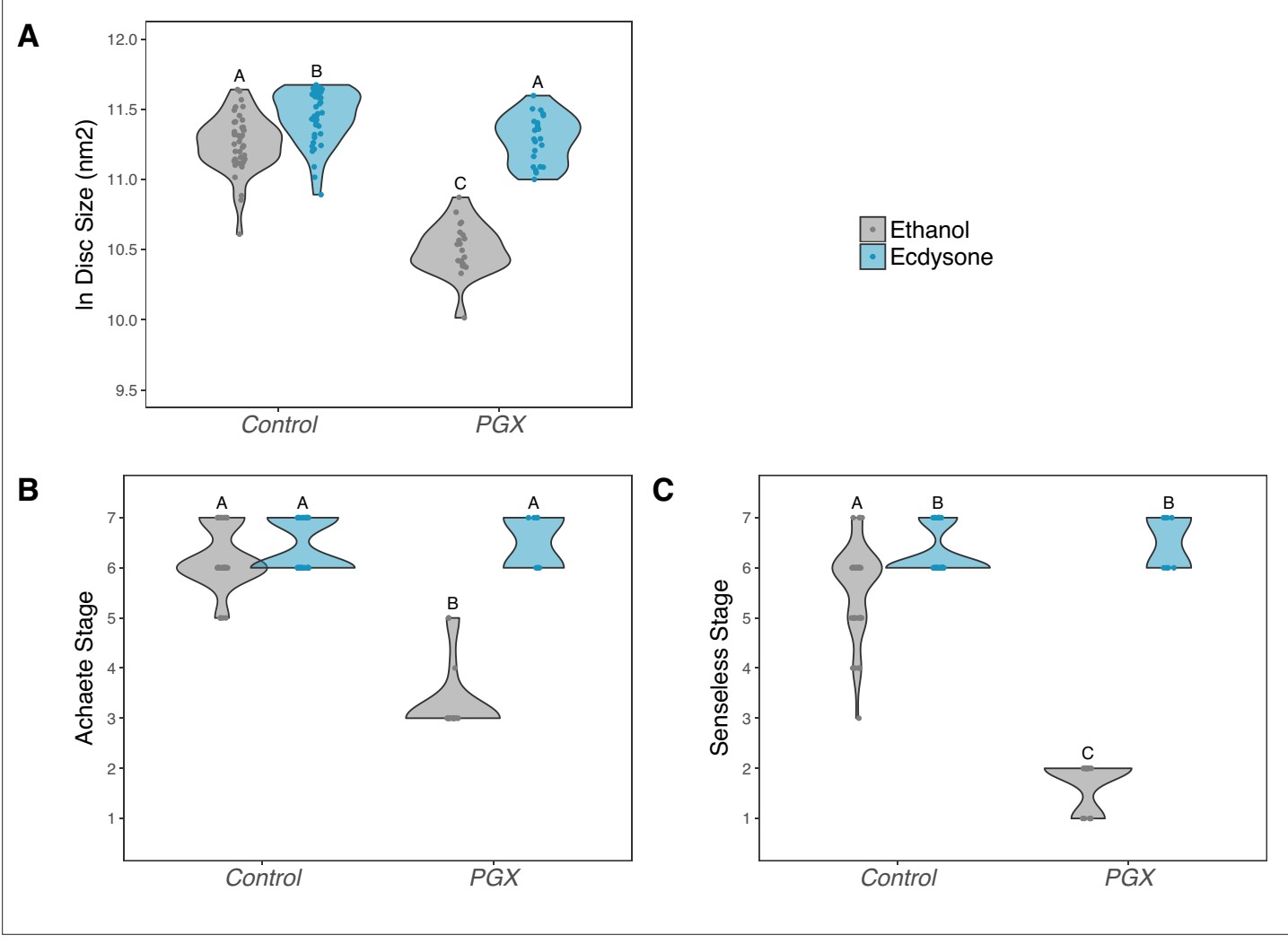

**Figure 4.** Supplementing genetically ablated prothoracic gland (PGX) larvae with 20-hydroxyecdysone (20E) rescues wing disc growth (**A**), and Achaete (**B**) and Senseless (**C**) patterning. Both the control and PGX larvae were exposed to 20E-treated food (0.15 mg/mg of food) or ethanol-treated food (which contains the same volume of ethanol) at 0 hr after the third instar moult. Wing discs were removed at 42 hr after the third instar moult. Control genotypes are the pooled results from both parental controls (either the *phm-GAL4; GAL80ts*, or *UAS-GRIM* parental strain crossed to w[1118]). Treatments marked with different letters are significantly different (Tukey's HSD, p<0.05, for ANOVA see **Supplementary file 1**d). Data were plotted using violin plots with individual wing discs displayed over the plots. $N_{PGX + ethanol}$ = 21, $N_{PGX + 20E}$ = 23, $N_{Control + ethanol}$ = 43, $N_{Control + 20E}$ = 42.

An important aspect of these data is that in PGX larvae either supplementing the 1% sucrose food with 20E or feeding them on normal food both rescued wing disc growth (**Figure 5A**), albeit incompletely. This suggests that nutrition can drive growth through mechanisms independent of ecdysone, and vice versa. In contrast, nutrition alone only marginally promoted Achaete and Senseless patterning in starved PGX larvae, while 20E alone completely restored patterning. Further, even early patterning did not progress in PGX larvae (**Figure 3**). Thus, the effect of nutrition on patterning appears to be wholly mediated by ecdysone, while the effect of nutrition on growth appears to be partially mediated by ecdysone and partially through another independent mechanism. Ecdysone-independent growth appears to occur early in the third larval instar, however, since disc growth in PGX and control larvae is more or less the same in the first 12 hr after ecdysis to L3 (**Figure 2**).

## Ecdysone drives growth and patterning independently

The data above suggest a model of growth and patterning, where both ecdysone and nutrition can drive growth, but where patterning is driven by ecdysone. We next focused on exploring how ecdysone regulates both growth and patterning. Patterning genes, particularly morphogens, are known to

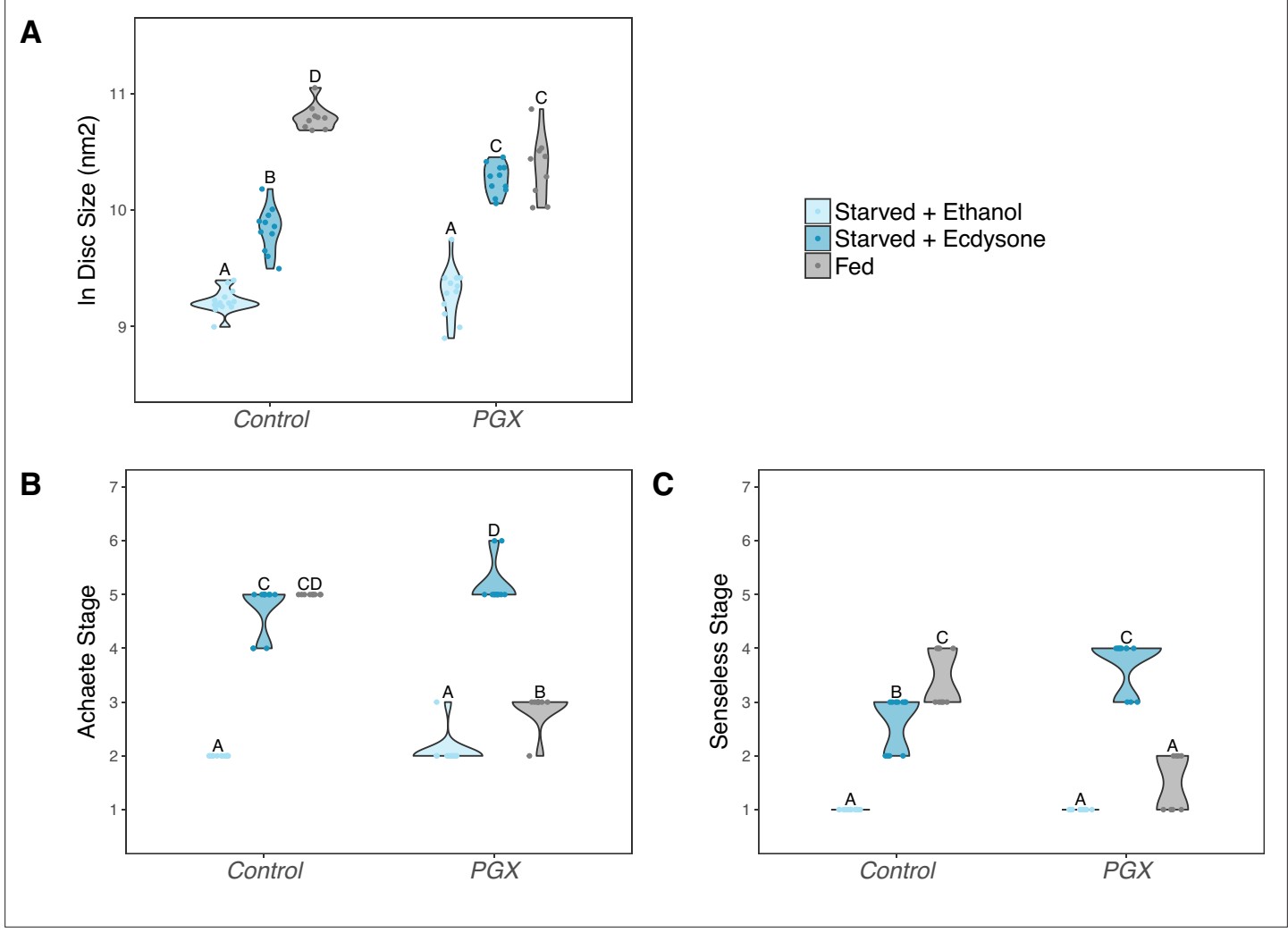

**Figure 5.** Supplementing genetically ablated prothoracic gland (PGX) larvae with 20-hydroxyecdysone (20E) is able to partially rescue the effect of yeast starvation on (**A**) wing discs growth, and fully rescue (**B**) Achaete and (**C**) Senseless patterning. Both the control and PGX larvae were exposed from 0 hr after the third instar moult to one of three food types: (1) starved + 20 E – starvation medium containing 1% sucrose and 1% agar laced with 20E (0.15 mg/mg of food), (2) starved + ethanol – starvation medium treated with the same volume of ethanol, or (3) fed – normal fly food. Wing discs were removed at 24 hr after the third instar moult. Control genotypes are the pooled results from both parental controls (the *UAS-GRIM* parental strain crossed to w[1118]). Treatments marked with different letters are significantly different (Tukey's HSD, p<0.05, for ANOVA see ***Supplementary file 1***e). Data were plotted using violin plots with individual wing discs displayed over the plots. $N_{PGX + starved - ethanol}$ = 23, $N_{PGX + starved - 20E}$ = 22, $N_{PGX + fed}$ = 26, $N_{Control + starved-ethanol}$ = 28, $N_{Control + starved-20E}$ = 22, $N_{Control + fed}$ = 27.

regulate growth, so one hypothesis is that ecdysone promotes patterning, which in turn promotes the ecdysone-driven component of disc growth. A second related hypothesis is that ecdysone-driven growth is necessary to promote patterning. Under either of these hypotheses, because the mechanisms regulating patterning and growth are interdependent, we would expect that changes in ecdysone levels would not change the relationship between disc size and disc pattern. An alternative hypothesis, therefore, is that ecdysone promotes growth and patterning through at least partially independent mechanisms. Under this hypothesis, the relationship between size and patterning may change at different levels of ecdysone.

To distinguish between these two hypotheses, we increased or decreased the activity of the insulin-signalling pathway in the PG, which is known to increase or decrease the level of circulating ecdysone, respectively (***Caldwell et al., 2005***; ***Koyama et al., 2014***; ***Colombani et al., 2005***; ***Mirth et al., 2005***). We then looked at how these manipulations affected the relationship between disc size and disc pattern, again focusing on Achaete and Senseless patterning. We increased insulin signalling in

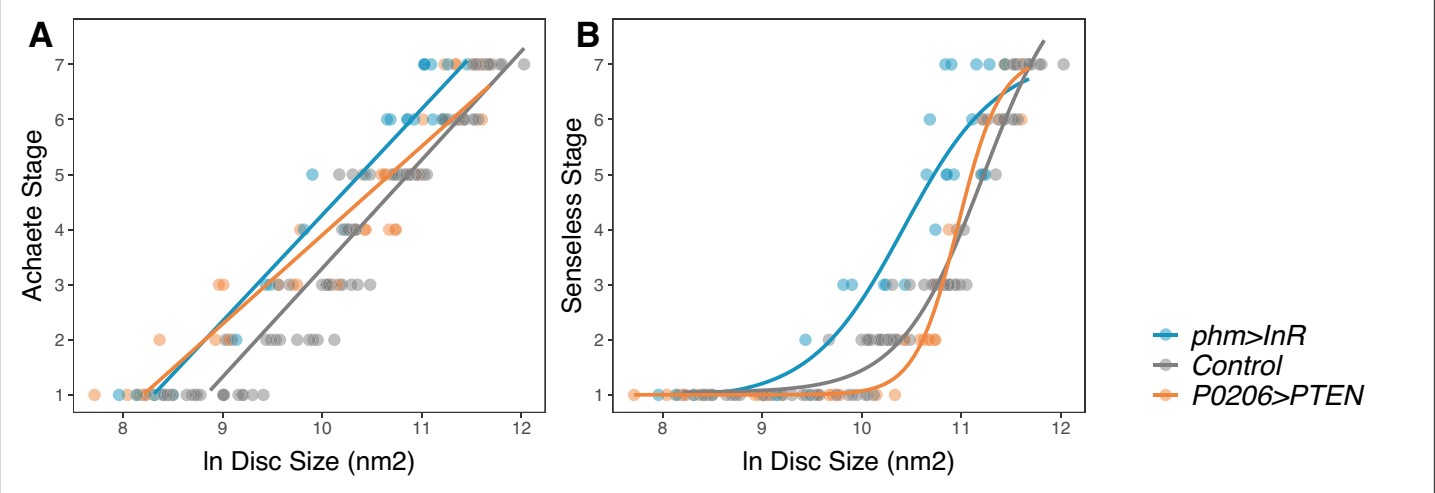

**Figure 6.** Changing rate of ecdysteroid synthesis changes the relationship between disc pattern and disc size throughout the third-instar larval stage. (**A**) The relationship between Achaete stage and disc size was fitted with a linear regression, the parameters of which are significantly different between genotypes (*Supplementary file 1*f). (**B**) The relationship between Senseless stage and disc size was fitted with a four-parameter logistic regression, the parameters of which are significantly different between genotypes (*Supplementary file 1*g). We staged third instar larvae from the onset of the moult to the formation of white prepupae. The length of this developmental interval varied across genotypes. For control genotypes, we sampled larvae at 0, 10, 20, 30, 48, and 51 (InR control)/53 (PTEN control) hr after third instar ecdysis (AL3E). For phm>InR, we sampled larvae at 0, 10, 20, 29, 30, and 36 hr AL3E. For P0206>PTEN, we sampled larvae at 0, 10, 20, 30, 40, 60, 73, and 80 hr AL3E. The number of discs sampled for each genotype and patterning gene was: for Achaete: $N_{phm>InR}$ = 29, $N_{Control}$ = 99, $N_{P0206>PTEN}$ = 40, for Senseless: $N_{phm>InR}$ = 29, $N_{Control}$ = 99, $N_{P0206>PTEN}$ = 41.

the PG by overexpressing InR (*phm>InR*) and reduced insulin signalling by overexpressing the negative regulator of insulin signalling PTEN (*P0206>PTEN*).

We found that a linear model is sufficient to capture the relationship between disc size and Achaete stage when we either increase (*phm>InR*: $AIC_{linear} - AIC_{logistic}$ = 22, ANOVA, $F_{(25,27)}$ = 1.71, p=0.2018) or decrease ecdysone synthesis rates (*P0206>PTEN*). Changing ecdysone levels, however, significantly changed the parameters of the linear model and altered the relationship between disc size and Achaete pattern. Specifically, increasing ecdysone level shifted the relationship so that later stages of Achaete patterning occurred in smaller discs (*Figure 6A*, *Supplementary file 1*f).

The relationship between Senseless pattern and disc size is best fit using a four-parameter logistic (threshold) function, which provides a significantly better fit to the data than a linear function ($AIC_{linear} - AIC_{logistic}$ = 32.2; ANOVA, $F_{(44,46)}$ = 25.8, p<0.001). Changing ecdysone levels significantly changed the parameters of the logistic model and altered the relationship between disc size and Senseless pattern (*Figure 6B*, *Supplementary file 1*g). Again, increasing ecdysone level shifted the relationship so that later stages of Senseless patterning occurred in smaller discs. Collectively, these data support the hypothesis that ecdysone acts on growth and patterning at least partially independently (*Figure 1D*; Hypothesis 3), and that patterning is not regulated by wing disc size (*Figure 1C*; Hypothesis 2).

## Ecdysone regulates disc growth and disc patterning through different mechanisms

The data above support a model whereby environmental signals act through ecdysone to co-regulate growth and patterning, generating organs of variable size but invariable pattern. Further, growth is also regulated by an ecdysone-independent mechanism, enabling similar progressions of pattern across discs of different sizes. An added nuance, however, is that ecdysone levels are not constant throughout development. Rather, the ecdysone titre fluctuates through a series of peaks throughout the third larval instar and the dynamics of these fluctuations are environmentally sensitive (*Warren et al., 2006*). To gain further insight into how ecdysone co-regulates plasticity and robustness, we therefore explored which aspects of ecdysone dynamics regulate growth and patterning.

Two characteristics of ecdysone fluctuations appear to be important with respect to growth and patterning. First, the timing of the ecdysone peaks set the pace of development, initiating key developmental transitions such as larval wandering and pupariation (*Koyama et al., 2014*; *Mirth et al.,*

*2005*; *Warren et al., 2006*; *Riddiford, 1993*). Second, the basal levels of ecdysone appear to regulate the rate of body growth, with an increase in basal level leading to a reduction in body growth (*Caldwell et al., 2005*; *Herboso et al., 2015*; *Colombani et al., 2005*; *Mirth et al., 2005*; *Mirth et al., 2014*). While several studies, including this one, have established that disc growth is positively regulated by ecdysone (*Herboso et al., 2015*; *Dye et al., 2017*; *Parker and Shingleton, 2011*), whether disc growth is driven by basal levels or peaks of ecdysone is unknown.

There are a number of hypotheses as to how ecdysone levels may drive patterning and growth. One hypothesis is that patterning and ecdysone-regulated disc growth show threshold responses, which are initiated once ecdysone rises above a certain level. This would manifest as low patterning and growth rates when ecdysteroid titres were sub-threshold, and a sharp, switch-like increase in patterning and growth rates after threshold ecdysone concentrations was reached. Alternatively, both may show a graded response, with patterning and growth rates increasing continuously with increasing ecdysteroid titres. Finally, disc patterning may show one type of response to ecdysone, while disc growth may show another. Separating these hypotheses requires the ability to titrate levels of ecdysone.

To do this, we reared PGX larvae on standard food supplemented with a range of 20E concentrations (0, 6.25, 12.5, 25, 50, and 100 ng of ecdysone/mg of food). However, as noted above, disc growth early in the third larval instar is only moderately affected by ablation of the PG, potentially obfuscating the effects of supplemental 20E. In contrast, discs from starved PGX larvae show no growth or patterning without supplemental 20E. We therefore also reared PGX larvae either on standard food or on 20% sucrose/1% agar medium (from hereon referred to as 'starved' larvae) supplemented with a range of 20E concentrations. For both control genotypes and PGX larvae, increasing the concentration of 20E in the food increased ecdysteroids titres in the larvae (*Figure 7—figure supplement 1*, *Supplementary file 1*h).

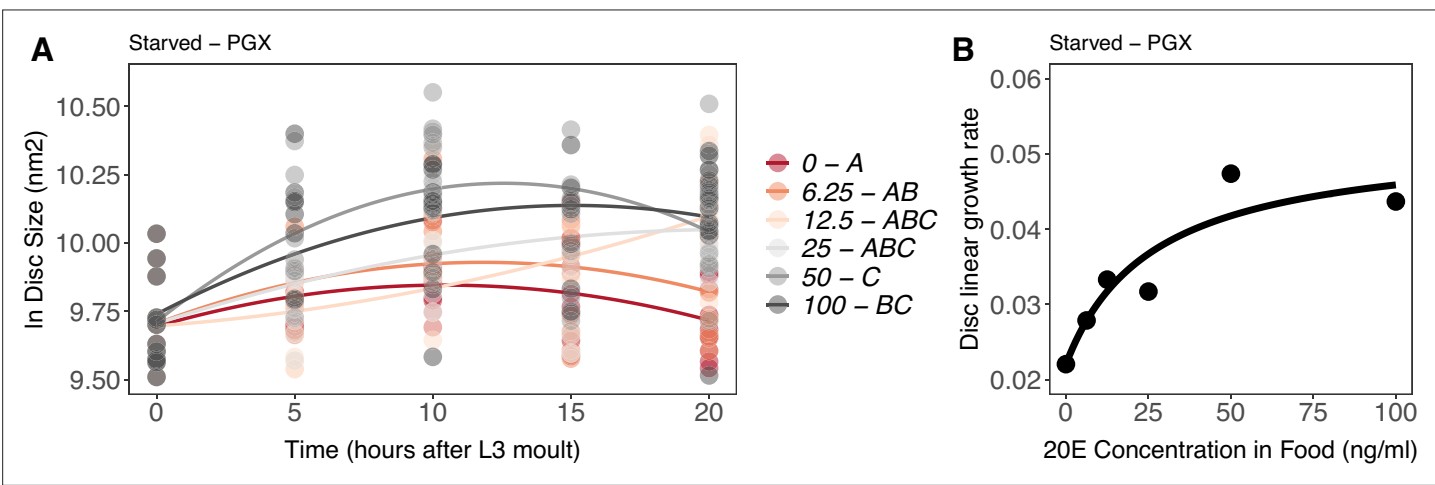

**Figure 7.** Effect of supplemental 20-hydroxyecdysone (20E) on growth of the wing imaginal disc in starved genetically ablated prothoracic gland (PGX) larvae. Growth was modelled as $S = E + T + T^2 + E * T + E * T^2$, where $S$ = disc size, $E$ = 20E concentration, and $T$ = disc age. (**A**) There was a significant effect of $E$ on the linear growth rate of the wing imaginal discs (*Supplementary file 1*j). Each point corresponds to a wing disc, $N_{PGX-starved}$ = 409 (63–72 discs were sampled per treatment across all time points). 20E treatments that do not share a letter (see legend) are significantly different in patterning rates as determined by post-hoc test on the slopes (*Supplementary file 1*j). (**B**) The linear growth rate of the wing disc was extracted from the growth model for each concentration of 20E and modelled using a three-parameter Michaelis–Menten equation: $y = c + (d-c)/(1 + (b/x))$, where $c$ is $y$ at $x = 0$, $d$ = $y[max]$, and $b$ is $x$ where $y$ is halfway between $c$ and $d$. Linear growth rate increases steadily with ecdysone concentration in the food up until 25 ng of 20E/ml of food, after which growth rate increases more slowly with increasing 20E concentration.

The online version of this article includes the following figure supplement(s) for figure 7:

**Figure supplement 1.** The effects of 20-hydroxyecdysone (20E) concentration in the food on ecdysteroid titres in control and genetically ablated prothoracic gland (PGX) larvae.

**Figure supplement 2.** Wing imaginal disc growth is suppressed in fed genetically ablated prothoracic gland (PGX) larvae relative to controls and in starved larvae of both genotypes.

**Figure supplement 3.** There is no effect of supplemental 20-hydroxyecdysone (20E) on growth of the wing imaginal disc in fed genetically ablated prothoracic gland (PGX) larvae.

To quantify the effects of 20E concentration in the wing disc growth, we dissected discs at 5 hr intervals starting immediately after the moult to the third instar (0 hr AL3E) to 20 hr AL3E. Because male and female larvae show differences in wing disc growth (**Testa et al., 2013**), we separated the sexes in this experiment and focused our analysis on female wing discs.

As before, in both PGX and control larvae, wing disc growth was suppressed by starvation (**Figure 7—figure supplement 2**, **Supplementary file 1**i). To explore how disc size changed over time with increasing 20E concentration, we modelled the data using a second-order polynomial regression against time after third instar ecdysis. Increasing the concentration of supplemental 20E increased the disc growth rate in starved PGX larvae (**Figure 7A**, **Supplementary file 1**j). In contrast, increasing 20E concentrations had no effect on disc growth rate in fed PGX larvae (**Figure 7—figure supplement 3**, **Supplementary file 1**j). This confirms that the effect of nutrition on growth masks the effect of 20E early in the third larval instar and supports that hypothesis that disc growth during this period is primarily regulated by nutrition and only moderately regulated by ecdysone (**Shingleton et al., 2008**).

To test whether wing disc growth rates show either a graded or threshold response to 20E concentration in starved PGX larvae, we extracted the linear growth rate coefficients from our models. We then modelled the relationship between growth rate and 20E concentration with three nonlinear functions: a graded Michaelis–Menten function, and threshold three- and four-parameter log-logistic functions. Finally, we tested which model best fit the data using Akaike information criteria (AIC) and Bayesian information criteria (BIC) for model selection. The model with the lowest AIC and BIC values best fits the data.

When wing disc growth rate was modelled with the graded Michaelis–Menten function, both the AIC and BIC values were lower than when it was modelled with either threshold function (**Supplementary file 1**k). This supports the hypothesis that growth rate increases continuously with increasing 20E concentration, with growth rate plateauing after 20E concentrations reach 25 ng/ml (**Figure 7B**). Thus, disc growth rate appears to show a graded response to 20E level in the absence of nutrition. This is in line with recent findings from **Strassburger et al., 2021**, which show that proliferation and growth in the wing discs increase with increasing 20E concentration in the diet (**Strassburger et al., 2021**).

The effect of 20E concentration on Achaete patterning was qualitatively different to its effect on growth. As before, Achaete patterning did not progress in either starved or fed PGX larvae (**Figure 8—figure supplement 1**). In contrast, Achaete patterning did progress in PGX larvae supplemented with 20E. Patterning rates for Achaete did not differ significantly between 0–6.25 ng/ml (fed) and 0–12.5 ng/ml (starved) of 20E (**Figure 8A and C**, **Supplementary file 1**). Above 25 ng/ml of 20E, Achaete patterning occurred at the same rapid rate in both fed and starved PGX larvae (**Figure 8A and C**).

To compare the progression of Achaete at different levels of 20E with the progression of disc growth, we modelled the relationship between Achaete pattern, time after third-instar ecdysis, and 20E concentration as second-order polynomial regression for fed and starved PGX larvae. As for disc growth, we then extracted the linear coefficients from this model at each level of 20E and modelled the relationship between patterning rate and 20E using a Michaelis–Menten function and a three- and four-parameter log-logistic function. Both the AIC and BIC indicated that that a threshold four-parameter log-logistic function fit the data better than a graded Michaelis–Menten function (**Supplementary file 1**k). Thus, unlike growth, Achaete patterning showed a threshold response to 20E concentration. Specifically, Achaete patterning was not initiated unless 20E is above a certain concentration (12.5–25 ng/ml), but it progressed at the same rate regardless of how high 20E is above this concentration.

Comparing the timing of Achaete patterning in 20E-supplemented PGX larvae versus fed controls provides some indication of when in normal development the threshold level of 20E necessary to initiate Achaete patterning is reached. Discs from fed control larvae began to reach Achaete stage 4 by 15 hr AEL3 (**Figure 8—figure supplement 1B**), while discs from both fed and starved PGX larvae supplemented with >25 ng/ml of 20E began to reach stage 4 by 10 hr AEL3 (**Figure 8A and C**). This suggests that in control larvae ecdysone levels sufficient to initiate Achaete patterning are only reached 15 hr after the moult to the third larval instar.

Senseless patterning did not progress as far as Achaete patterning, only achieving an average of stage 3 in fed and stage 4 in starved larvae at 20 hr AL3E when supplemented with 20E. In both fed and starved larvae, supplemental 20E at or below 12.5 ng/ml was insufficient to rescue Senseless

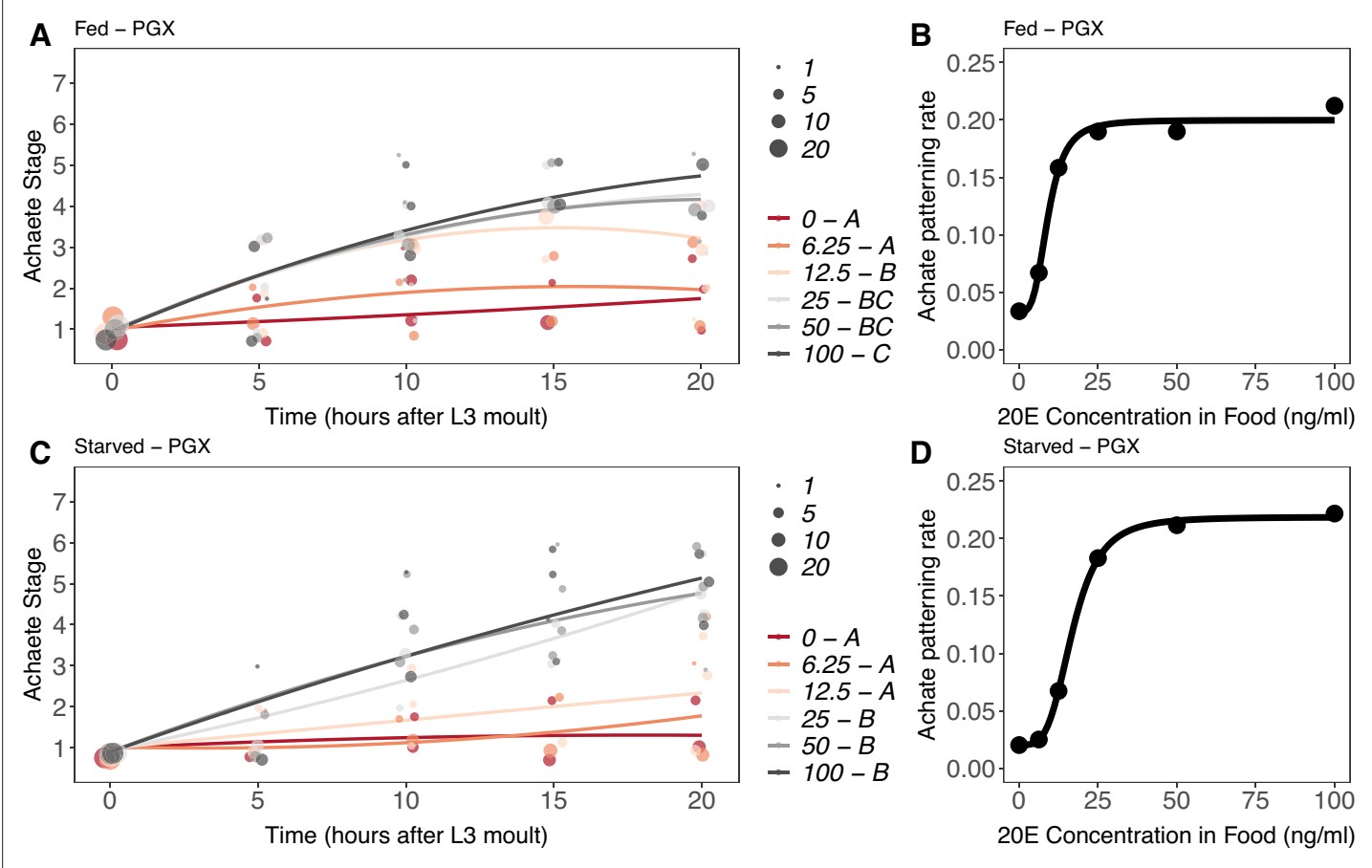

**Figure 8.** Effect of supplemental 20-hydroxyecdysone (20E) on Achaete patterning of the wing imaginal disc in (**A, B**) fed and (**C, D**) starved genetically ablated prothoracic gland (PGX) larvae. In (**A**) and (**C**), patterning stage was modelled as $A = E + T + T^2 + E * T + E * T^2$, where $A$ = Achaete stage, $E$ = 20E concentration, and $T$ = disc age. The size of each point corresponds to the number of wing disc in each stage. 20E treatments that do not share a letter (see legend) are significantly different in patterning rates as determined by post-hoc test on the slopes (for ANOVA, see **Supplementary file 1**l). In (**B**) and (**D**), we extracted the linear patterning rate from fed (**B**) or starved (**D**) PGX larvae. We then modelled the relationship between patterning rate and 20E concentration using a four-parameter log-logistic equation: $y = c + (d)(-c)/(1 + e^{(b(log(x))-log(a))})$, where $c$ is the lower asymptote, $d$ is the upper asymptote, $b$ is the rate of increase, and $a$ is the inflection point. $N_{PGX-fed}$ = 459, $N_{PGX-starved}$ = 409, 63–86 discs were sampled per treatment across all time points.

The online version of this article includes the following figure supplement(s) for figure 8:

**Figure supplement 1.** Achaete patterning in wing discs from fed and starved genetically ablated prothoracic gland (PGX) and control larvae.

patterning, while supplemental 20E at or above 25 ng/ml rescued patterning to approximately the same extent (**Figure 9A and C**, **Supplementary file 1**m). As for Achaete patterning, supplemental 20E also initiated Senseless patterning in PGX larvae early when compared to fed controls. Discs from fed control larvae began to reach Senseless stage 3 at 20 hr AEL3 (**Figure 9—figure supplement 1**), while discs from both fed and starved PGX larvae supplemented with ≥25 ng/ml of 20E were at stage 3 by 15 hr AEL3 (**Figure 9A and C**).

We again used a second-order polynomial regression to model the relationship between Senseless pattern, time after third-instar ecdysis, and 20E concentration. The relationship between the linear coefficient from our model for Senseless patterning and 20E was best fit with a log-logistic rather than a Michaelis–Menten function. For fed PGX larvae, the four-parameter logistic function provided the best fit to the data (**Figure 9B**, **Supplementary file 1**k), whereas for starved PGX larvae, the three-parameter logistic function was the best fit (**Figure 9D**, **Supplementary file 1**k). Thus, like Achaete patterning rate, Senseless patterning rate showed a threshold responses to 20E concentration.

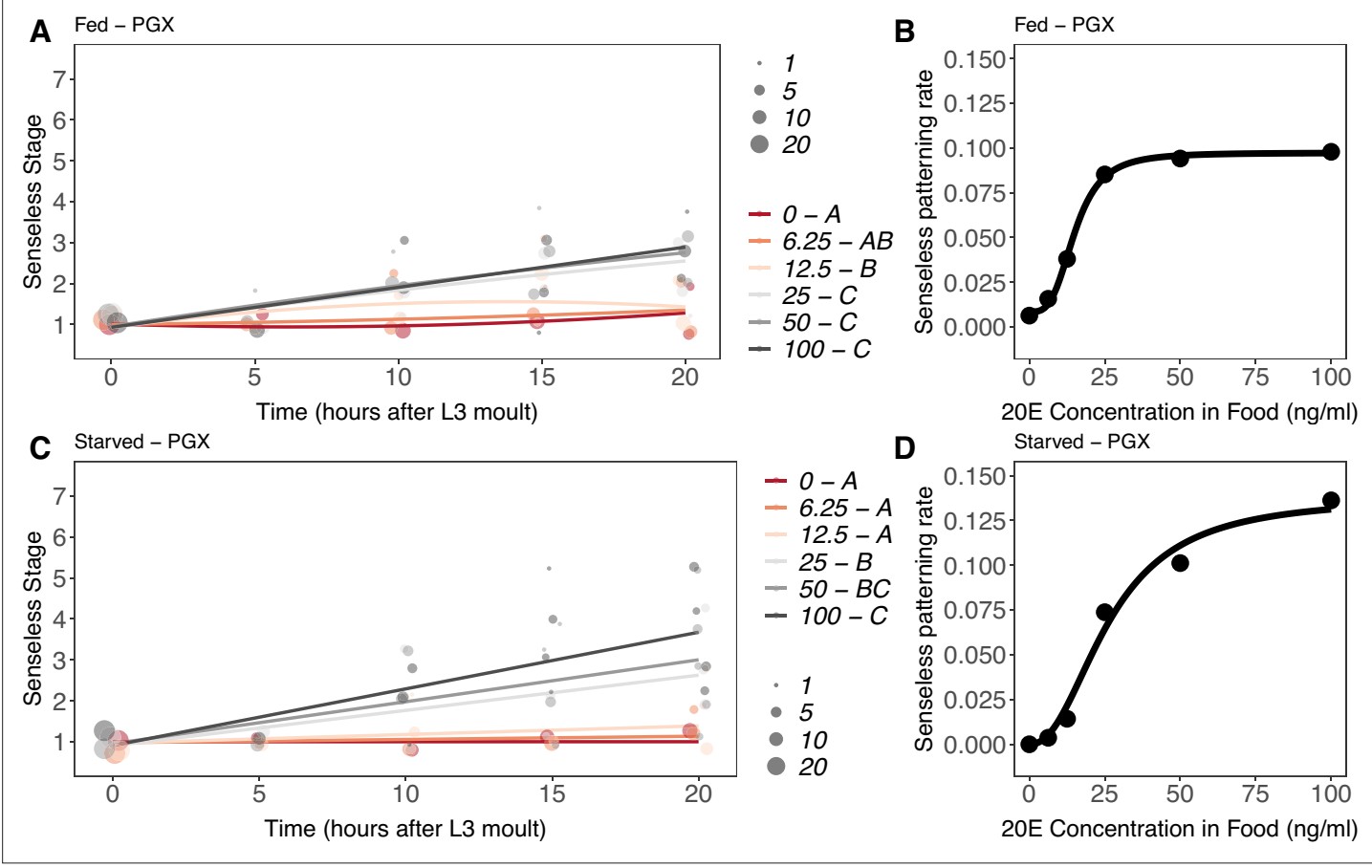

**Figure 9.** Effect of supplemental 20-hydroxyecdysone (20E) on Senseless patterning of the wing imaginal disc in (**A, B**) fed and (**C, D**) starved genetically ablated prothoracic gland (PGX) larvae. In (**A**) and (**C**), patterning stage was modelled as $S = E + T + T^2 + E * T + E * T^2$, where $S$ = Senseless stage, $E$ = 20E concentration, and $T$ = disc age. The size of each point corresponds to the number of wing discs in each stage. 20E treatments that do not share a letter (see legend) are significantly different in patterning rates as determined by post-hoc test on the slopes (for ANOVA, see *Supplementary file 1*m). In (**B**) and (**D**), we extracted the linear patterning rate in fed (**B**) or starved (**D**) larvae. We then modelled the relationship between patterning rate and 20E concentration in (**B**) using a four-parameter log-logistic equation: $y = c + (d)(-c)/(1 + e^{(b(log(x))-log(a))})$, where $c$ is the lower asymptote, $d$ is the upper asymptote, $b$ is the rate of increase, and $a$ is the inflection point. In (**D**), we used a three-parameter log-logistic equation: $y = d/(1 + e^{(b(log(x))-log(a))})$, where $d$ is the upper asymptote, $b$ is the rate of increase, and $a$ is the inflection point. $N_{PGX-fed}$ = 459, $N_{PGX-starved}$ = 409, 63–86 discs were sampled per treatment across all time points.

The online version of this article includes the following figure supplement(s) for figure 9:

**Figure supplement 1.** Senseless patterning in wing discs from fed and starved genetically ablated prothoracic gland (PGX) and control larvae.

Collectively, these data support a model of disc growth and patterning where ecdysone regulates disc growth as a graded response to basal levels of ecdysone, while ecdysone regulates disc patterning as a single threshold response.

## Discussion

Organs are remarkably good at achieving correct pattern across a broad range of environmental conditions that generate variation in size. While this might seem a simple feat when growth and patterning occur at separate times or are regulated by different hormones, it is considerably less simple if both growth and patterning occur at the same time and are regulated by the same endocrine signal. In this work, we explored how the wing discs of developing *D. melanogaster* use the same hormonal signal to coordinate both their growth and progression of pattern. We found that ecdysone simultaneously regulates the plastic growth and robust patterning of the wing disc through independent mechanisms: plastic growth responds to ecdysone with a graded response, while robust

patterning responds with a single threshold response. We propose that these differences in response represent a potentially general mechanism through which high levels of variation in one organ characteristic, for example, size, could be coordinated with low levels of variation in another characteristic of the same organ, for example, pattern.

These data make an important contribution to our understanding of how environmental factors, specifically nutrition, affect growth and patterning in developing organs in *Drosophila*. During normal development, circulating ecdysone levels are low during the first 8 hr of the third larval instar until attainment of a critical size initiates a hormonal cascade that causes ecdysone to fluctuate through a series of characteristic peaks. Each of these peaks is associated with key developmental transitions and prepares the larva for metamorphosis. Low nutrition delays attainment of critical size and the initiation of these peaks, but also appears to raise basal levels of circulating ecdysone between these peaks (*Lee et al., 2018*), which slows the growth of the body (*Lee et al., 2018*). At the same time, low nutrition also lowers the levels of circulating insulin-like peptides, further slowing the growth of the body. While low levels of insulin signalling will suppress imaginal disc growth, the increase in ecdysone concentrations resulting from starvation opposes some of these effects (*Herboso et al., 2015*; *Dye et al., 2017*; *Parker and Shingleton, 2011*) by promoting imaginal disc growth.

Our data suggest that these opposing effects are critical to robust patterning of the wing under different nutritional conditions. At low nutritional conditions, low insulin signalling at the beginning of the third larval instar slows the growth of the body and the imaginal discs. At this stage, growth of the wing imaginal discs is less dependent on ecdysone (*Shingleton et al., 2008*), evident from the more moderate effects on growth of the wing discs during this period in fed PGX larvae. In the middle of the third larval instar, however, low nutritional conditions elevate basal levels of ecdysone (*Lee et al., 2018*). This drives disc growth independent of insulin signalling to ensure the discs are of sufficient size to generate viable adult appendages, even as elevated ecdysone suppresses the growth of the body as a whole (*Caldwell et al., 2005*; *Colombani et al., 2005*; *Mirth et al., 2005*;). At the same time, changes in the tempo of the ecdysone fluctuations may ensure that patterning is initiated at the appropriate developmental time, when discs are sufficient size to generate a viable adult appendage. Three factors therefore appear necessary to achieve variable size but robust patterning under a range of nutritional conditions: (1) a graded growth response to ecdysone, (2) nutritionally sensitive growth that is independent of ecdysone, and (3) a threshold patterning response to ecdysone.

There is some evidence that our findings apply to patterning and growth of the wings in other insect species. In the tobacco hornworm *Manduca sexta* and the buckeye butterfly *Junonia coenia*, wing disc growth is regulated by both ecdysone and insulin (*Strassburger et al., 2021*; *Parker and Struhl, 2015*). In the butterfly *J. coenia*, the patterning stage of wing discs can be quantified by the extent of tracheal invasion, resulting in wing vein patterning (*Miner et al., 2000*). In this species, wing vein patterning progresses independently of wing size in starved versus fed caterpillars (*Miner et al., 2000*). Thus, the independent regulation of growth and patterning, with growth regulated by both insulin and ecdysone signalling, may be a general mechanism to achieve robust patterning across a range of wing sizes.

While ecdysone and insulin signalling provide systemic cues that tune organ growth to the environmental conditions, morphogens like Wingless and Decapentaplegic (Dpp) act to regulate growth in an organ-autonomous manner. The extent to which morphogen gradients respond to these systemic cues is unclear, although the activity of morphogens is known to interact with those of systemic signals at the level of the target genes. For example, insulin/TOR signalling regulates the activity of Yorkie, a downstream effector of patterning morphogens, including Wingless and Dpp, which controls the rate of cell division (*Parker and Struhl, 2015*). Similarly, reducing ecdysone signalling in the wing reduces the expression of Wingless and reduces Dpp signalling, measured by the levels of phosphorylated Mothers against Dpp expression (*Herboso et al., 2015*; *Dye et al., 2017*; *Mirth et al., 2009*). Taken together, the signalling pathways that regulate organ growth in response to environmental conditions interact in complex ways with those that regulate organ-autonomous growth, suggesting that these two growth-regulating mechanisms are not as independent as previously thought (*Mirth and Shingleton, 2019*).

Although the growth of the disc relies on insulin and ecdysone signalling, the progression of patterning for Achaete and Senseless in the wing disc appears to be driven by threshold responses to ecdysone. This is not to say that the progression of patterning does not depend on environmental

conditions. Indeed, starvation early in the third instar impedes patterning in both the wing and ovary of *D. melanogaster* (*Mirth et al., 2009*; *Mendes and Mirth, 2016*). However, rather than resulting from a direct effect of insulin signalling on patterning, the block in the progression of pattern occurs because insulin signalling controls the timing of the first ecdysone pulse in the third larval instar (*Koyama et al., 2014*; *Ohhara et al., 2017*). Our results here confirm that patterning requires suprathreshold concentrations of ecdysone to be initiated. Further, the manner in which ecdysone regulates the progression of patterning ensures that it remains robust against further environmental perturbation. By switching on pattern above threshold ecdysone concentrations, the disc can continue to pattern across a range of environmental conditions, even while growth retains sensitivity to those conditions.

A similar threshold mechanism appears to regulate patterning in the wing discs of other insects. As for *Drosophila,* the earliest stages of wing patterning depend on nutrition in *J. coenia*. If caterpillars are starved before the wing discs begin to pattern, then their discs remain small and their veins unpatterned (*Miner et al., 2000*). In caterpillars starved at later stages after disc patterning has been initiated, the wing discs are small but reach the same vein patterning stage as those of fed control animals. Whether or not the initiation of patterning in *J. coenia* also depends on ecdysone has yet to be determined.

At first glance, the observation that patterning shows a threshold response to ecdysone may not be surprising. In any given cell, patterning is inherently regulated by threshold responses because the expression of the patterning gene product is either on or off in that cell. However, our patterning scheme considers the progression of patterning across the entire field of cells that make up the wing disc. Cells across the wing disc turn on Achaete and Senseless expression at different times, resulting in a continuous progression of pattern with time (*Oliveira et al., 2014*). Furthermore, like growth, the progression of pattern can vary in rate depending on environmental and hormonal conditions (*Oliveira et al., 2014*). Consequently, the progression of patterning could, in principle, also show a graded response to ecdysone levels. The observation that once ecdysone concentrations are above threshold, the rate of patterning for Achaete and Senseless is independent of ecdysone provides evidence that the rate of patterning across an entire organ can also show a threshold response: an assumption that, hitherto, has not been tested.

What determines how progression of pattering unfolds through time is unclear. We did not observe discs progressing from stage 1 immediately to stage 7 within a single 5 hr time interval even at the highest 20E concentrations. This suggests that there are additional temporal factors that regulate the order of patterning progression. Almost certainly, interactions between the gene regulatory networks that regulate patterning control how patterning progresses across regions of the wing disc. We have very little understanding if/how the different regions of the wing communicate with each other to achieve this. In principle, differences between when cells turn on Achaete and Senseless across the disc could arise in response to other developmental signals, such as from the Dpp, Wingless, or Hedgehog morphogen gradients responsible for correctly scaling and patterning the wing.

Part of this temporal signature might arise from ecdysone itself. In this study, we exposed animals to tonic concentrations of ecdysone. Developing larvae, however, secrete four pulses of ecdysteroids between the moult to the third instar and pupariation (*Warren et al., 2006*). We have little understanding of how developmental information is encoded within these pulses. In principle, individual pulses could either prime tissues to become responsive to hormones or could alter their sensitivity – as the early ecdysone pulse does for wing disc growth and patterning (*Mirth et al., 2009*; *Mendes and Mirth, 2016*; *Shingleton et al., 2008*). Future studies comparing the difference between tonic and phasic exposure to hormone would help clarify the roles of the ecdysone pulses.

While our study has focussed on contrasting the robustness of patterning with plasticity of growth, depending on what is being measured there are instances where we expect patterning to also show plasticity (*Mirth et al., 2021*). For example, although the specification of cell types in the correct location within an organ may show little variation across environmental conditions, the number of structures specified can vary. The total number of abdominal and sternopleural bristles varies with temperature (*Moreteau and David, 2005*; *Moreteau et al., 2003*), as does the number of terminal filament stacks that are specified in the ovary, which is also affected by nutrition (*David, 1970*; *Delpuech et al., 1995*; *Green and Extavour, 2014*; *Hodin and Riddiford, 2000*). Plasticity in the number of bristle cells or terminal filament stacks presumably occurs because the mechanisms that specify the number of each structure do not scale with organ size. In other cases, the location of specific cell types may also be

plastic. For example, there is extensive literature exploring how the relative positions of veins in the wings of *D. melanogaster* and other insects are affected by environmental factors such as nutrition and temperature (e.g., *Debat et al., 2003*; *Debat et al., 2009*; *Outomuro et al., 2013*; *Bitner-Mathé and Klaczko, 1999*). Plasticity in wing shape is likely to be more complex and may involve a process that acts at many different points during wing development (*Matamoro-Vidal et al., 2015*; *Cobham and Mirth, 2020*). Future studies targeting how the mechanisms that establish the position of cell types differ from those that determine the number of cells of a given type would allow us to further define what makes traits either sensitive or robust towards changes in environmental conditions, and at what level.

## Materials and methods

### Fly stocks and rearing conditions

We manipulated growth rates and developmental timing by altering the rates of ecdysone synthesis in developing *D. melanogaster* larvae. To accelerate the rates of ecdysone synthesis, we used the progeny from $w^{1118}$;*phantom*-GAL4, which is expressed in the PGs, crossed with yw *flp; UAS InR29.4* (*phm>InR*). We decreased rates of ecdysone synthesis by crossing *P0206-GAL4*, which drives expression throughout the ring gland, with *yw; UAS PTEN* (*P0206>PTEN*). Even though *P0206*-GAL4 is a weaker GAL4 driver for the PG and also drives expression in the corpora allata, we chose to use it to drive *UAS PTEN* because *phm>PTEN* larvae die as first-instar larvae (*Mirth et al., 2005*). The parental lines *yw flp; UAS InR29.4* (*+>InR*) and *yw; UAS PTEN* (*+>PTEN*) were used as a reference for the *phm>InR* and *P0206>PTEN* genotypes, respectively.

Flies of the above genotypes were raised from timed egg collections (2–6 hr) on cornmeal/molasses medium containing 45 g of molasses, 75 g of sucrose, 70 g of cornmeal, 10 g of agar, 1100 ml of water, and 25 ml of a 10% Nipagin solution per litre. Larvae were reared at low density (200 eggs per 60 × 15 mm Petri dish) in a 12 hr light-dark cycle with 70% humidity and maintained at 25°C unless stated otherwise.

We used a transgenic combination that allowed us to genetically ablate the PG and eliminate native ecdysone synthesis specifically in the third larval instar. We crossed a *tub-GAL80*ts, *phantom GAL4* strain with *UAS Grim* to generate *PGX* progeny (*Herboso et al., 2015*). GAL80ts is a repressor of GAL4 active at temperatures lower than 22°C (*McGuire et al., 2003*). Rearing PGX larvae at 17°C allows GAL80ts to remain active, thus the *phantom GAL4* cannot drive the expression of *UAS grim* to promote cell death. Under these conditions, larvae can moult, pupariate, and complete metamorphosis (*Herboso et al., 2015*). Changing the larval rearing temperature to 29°C disables GAL80ts activity, thus ablating the PG (*Herboso et al., 2015*). The progeny of the inbred control strain, *w1118*, crossed with one of two parental lines, either *phantom*-GAL4 (*PG>+*) or *UAS Grim* (*+>Grim*), were used as controls for genetic background effects. The parental controls were reared under the same thermal conditions as PGX larvae.

Crosses, egg collections, and larval rearing were done on the cornmeal/molasses medium (above) for the experiments in *Figures 2–6* or, for the experiments in *Figures 7–9*, on Sugar-Yeast-Agar (SYA) medium: 50 g of autolysed Brewer's yeast powder (MP Biomedicals), 100 g of sugar, 10 g of agar, and 1200 ml of water. In addition, we added 3 ml of proprionic acid and 3 g of nipagen to the SYA medium to prevent bacterial and fungal growth. Egg collections were performed on SYA medium for 4 hr at 25°C or overnight at 17°C and larvae were reared at controlled densities of 200 eggs per food plate (60 × 15 mm Petri dish filled with SYA medium) at 17°C, as described previously (*Herboso et al., 2015*).

### Animal staging and developmental time

To measure the effects of changes in the rates in ecdysone synthesis on wing disc growth and wing disc patterning, larvae were staged into 1 hr cohorts at ecdysis to the third larval instar as in *Mirth et al., 2005* and *Mirth et al., 2009*. To do this, food plates were flooded with 20% sucrose and all second-instar larvae were transferred to a new food plate. After 1 hr, the food plate was flooded once again with 20% sucrose and the newly moulted third-instar larvae were collected and transferred to new food plates and left to grow until the desired time interval. Animals were staged and their wing discs dissected at defined intervals after the larval moult as in *Oliveira et al., 2014*.

For the experiments in *Figures 7–9*, PGX, *phm>*, and *>Grim* genotypes, larvae were raised from egg to second instar at 17°C. Larvae were staged into 2 hr cohorts at ecdysis to the third larval instar using the methods described above. We separated female and male larvae by examining them for the presence of testes, which are significantly larger than the ovaries and visible even in newly moulted males.

## Exogenous ecdysone feeding treatments

To show that ecdysone could rescue patterning and growth in PGX larvae (*Figure 4C and D*), we added either 0.15 mg of 20E (Cayman Chemical, item no. 16145) dissolved in ethanol, or an equivalent volume of ethanol, to 1 ml of standard food. Both the ethanol- and ecdysone-supplemented food were allowed to sit at room temperature for at least 4 hr to evaporate excess ethanol before use. Twelve larvae were transferred to one of the two supplemented foods either at 0 hr AL3E and left to feed for 42 hr or at 42 hr AL3E and left to feed for 24 hr.

To determine the relative contributions of nutrition-dependent signalling or ecdysone to growth and patterning, we fed newly moulted PGX and control larvae 1 ml of starvation medium (1% sucrose with 1% agar) supplemented with either 0.15 mg of 20E dissolved in ethanol or an equivalent volume of ethanol (Figure S4). Supplemented food was left at room temperature for at least 4 hr to evaporate excess off ethanol before use. Larvae were collected at 24 hr AL3E for tissue dissection.

For the 20E dose–response experiments, we conducted an initial pilot that showed that supplementing the food with 100 ng of ecdysone/mg food could rescue most of the Achaete and Senseless patterning in PGX wing discs. We collected newly moulted third instar larvae, separated the sexes, and then transferred 10–20 larvae to either sucrose food (20% sucrose, 1% agar; starved) or SYA food (fed) at 29°C. We fed these larvae on one of six 20E concentrations: 0, 6.25, 12.5, 25, 50, or 100 ng of 20E/mg food. We added the same volume of ethanol to all treatments.

To quantify the relationship between the concentration of 20E administered and the concentration of ecdysteroids in the hemolymph, we allowed newly ecdysed larvae to feed on either sucrose or SYA food that had been supplemented with one of the six concentrations of 20E for 20 hr at 29°C. We then transferred them onto either sucrose food or SYA food that did not contain ecdysone but was dyed blue. They were left to feed for 2 hr until their guts were filled with blue food. This extra step was taken so that we could be sure that our hemolymph ecdysone titres were not contaminated with ecdysone from the food. 30–40 larvae were then weighed as a group and transferred to five times their weight in volume of ice-cold methanol. Larvae were homogenized and ecdysone titres were determined using a 20-Hydroxyecdysone Enzyme ImmunoAssay Kit (Cayman Chemical, item no. 501390) as per the manufacturer's instructions.

## Dissections and immunocytochemistry

For each sample, 10–20 larvae were dissected on ice-cold phosphate-buffered saline (PBS) and fixed in 4% formaldehyde in PBS overnight at 4°C. After fixation, the tissue was washed four times (15 min per wash) with 0.3% Triton X-100 in PBS (PBT), then blocked for 30 min at room temperature in 2% heat-inactivated normal donkey serum in PBT. After blocking, the tissue was incubated in a primary antibody solution diluted with 2% heat-inactivated normal donkey serum in PBT overnight at 4°C. We used the guinea pig anti-Senseless (*Nolo et al., 2000*, 1:1000) and mouse anti-Achaete (Developmental Studies Hybridoma Bank, contributor J. Skeath, supernatant, 1:10) primary antibodies. To compare signal across tissues, we stained for both antigens simultaneously. The washing and blocking procedure was repeated after primary antibody incubation, and then the tissue was incubated in a secondary antibody (1:200 each of anti-guinea pig [Alexa Fluor 546] and anti-mouse [Alexa Fluor 488]) overnight at 4°C. The tissues were washed with PBT and rinsed with PBS, and then the wing imaginal discs were mounted on poly-L-lysine-coated coverslips using Fluoromount-G (SouthernBiotech). Tissues were imaged using either a Leica LSM 510 or a Nikon C1 upright confocal microscope and processed using ImageJ (version 2.0) and Adobe Photoshop CC 2017.

## Quantifications of wing imaginal disc size and Achaete and Senseless pattern

We quantified wing disc size using disc area as a proxy. All quantifications were done using ImageJ. Wing discs show exponential growth in the third instar. Thus, we studied the growth trajectories of the discs by ln-transforming disc area.

Achaete and Senseless stage was quantified using the staging scheme developed by *Oliveira et al., 2014*, associating each of the wing imaginal discs to an Achaete or Senseless stage varying from 1 to 7.

## Statistical analysis

All the analyses were conducted in R and the annotated R markdown scripts, and data for the analyses are deposited on Figshare (doi: 10.26180/13393676).

For the relationship between time after third-instar ecdysis and disc size (log µm²) or disc pattern (Achaete or Senseless), we fit either linear or Gompertz models and selected the model that best fit the data using ANOVA and AIC. The Gompertz model was parameterized as $y = ae^{-b*c^x}$, where $y$ is disc size/pattern, $x$ is time, $a$ is the asymptote of $y$, $b$ controls where along the x-axis the curve is positioned, and $c$ is the scaling constant, such that $c = e^g$, where $g$ is the growth/patterning rate (thus, the higher $g$ the lower $c$). To compare the parameters of linear models between treatments and genotypes, we used ANOVA. To compare the parameters of Gompertz models between treatments and genotypes, we used ANOVA to compare the fit of models that assign the same constants across groups versus models that assigned group-specific constants.

For the relationship between disc size (log µm²) and Senseless pattern, we fit a four-parameter logistic model parametrized as $y = c + \frac{(d-c)}{1+e^{(b-x)/a}}$ where $y$ is disc pattern, $x$ is disc size, $c$ is the minimum asymptote, $d$ is the maximum asymptote, $b$ is the inflection point, and $a$ is the scaling constant, such that $a = 1/k$, where $k$ is the logistic growth rate. We again used ANOVA to compare the fit of models that assign the same parameters across groups versus models that assigned group-specific parameters. The relationship between disc size and Achaete pattern was fit using a linear model and compared across treatments using ANOVA.

We used ANOVA to compare disc size/pattern at specific time points between treatments and genotypes using a Tukey's HSD test to allow comparison among groups.

Finally, to compare the effects of 20E supplementation in the diet on the progression of wing disc growth, Achaete patterning, and Senseless patterning, we fit a second-order orthogonal polynomial regression using disc size/patterning stage as our dependent variable, and 20E concentration and linear and quadratic terms for time as fixed effects. Fitting a single model to the data allowed us to compare the same model parameters for growth and patterning. We then extracted the linear rate of change at each 20E concentration using the *emtrends* function of the *emmeans* package in R (*Lenth, 2020*). The changes in growth/patterning rate with 20E concentration were modelled using three nonlinear functions: (1) a continuous Michaelis–Menten function: $y = c + \frac{(d-c)}{1+\frac{b}{x}}$, where $c$ is $y$ at $x = 0$, $d$ is the maximum asymptote, and $b$ is $x$ where $y$ is halfway between $c$ and $d$; (2) a threshold three-parameter log-logistic function: $y = \frac{d}{1+e^{b(\log x - \log a)}}$, where $d$ is the maximum asymptote, $b$ is the rate of increase, and $a$ is the inflection point; and (3) a threshold four-parameter log-logistic function: $y = c + \frac{(d-c)}{1+e^{b(\log x - \log a)}}$, where $c$ is the minimum asymptote, $d$ is the maximum asymptote, $b$ is the rate of increase, and $a$ is the inflection point. For each model, we calculated the AIC and BIC to allow model selection. The model that produces the lowest AIC and BIC value best fits the data.

For all parametric tests, we checked for homoscedasticity and normality of errors.

## Acknowledgements

We thank the members of the Mirth and Shingleton labs, past and present, for their useful discussions relating to this project. We also thank the staff at the Instituto Gulbenkian de Ciência - Advanced Imaging Lab and at Monash MicroImaging (MMI) for their technical support. This research was supported by NSF grants IOS-0919855, IOS-1557638, and IOS-1952385 to AWS and an ARC Future Fellowship (FT170100259) to CKM.

## Additional information

### Funding

| Funder | Grant reference number | Author |
|---|---|---|
| Australian Research Council | FT170100259 | Christen K Mirth |
| National Science Foundation | IOS-0919855 | Alexander Shingleton |
| National Science Foundation | IOS-1557638 | Alexander Shingleton |
| National Science Foundation | IOS-1952385 | Alexander Shingleton |

The funders had no role in study design, data collection and interpretation, or the decision to submit the work for publication.

### Author contributions

André Nogueira Alves, Alexander Shingleton, Formal analysis, Investigation, Methodology, Writing – original draft; Marisa Mateus Oliveira, Conceptualization, Formal analysis, Investigation, Methodology, Writing – original draft; Takashi Koyama, Investigation, Methodology; Christen Kerry Mirth, Conceptualization, Formal analysis, Funding acquisition, Investigation, Methodology, Project administration, Supervision, Visualization, Writing – original draft, Writing – review and editing

### Author ORCIDs

Takashi Koyama ![ORCID] http://orcid.org/0000-0003-4203-114X
Alexander Shingleton ![ORCID] http://orcid.org/0000-0001-9862-9947
Christen Kerry Mirth ![ORCID] http://orcid.org/0000-0002-9765-4021

### Decision letter and Author response

Decision letter https://doi.org/10.7554/eLife.72666.sa1
Author response https://doi.org/10.7554/eLife.72666.sa2

## Additional files

### Supplementary files
• Transparent reporting form
• Supplementary file 1. Tables - Results of Statistical Analyses.

### Data availability

All data and R scripts for analysis have been deposited on Figshare (https://doi.org/10.26180/13393676).

The following dataset was generated:

| Author(s) | Year | Dataset title | Dataset URL | Database and Identifier |
|---|---|---|---|---|
| Alves AN, Oliveira AM, Koyama T, Shingleton AW, Mirth C | 2020 | Dataset for Nogueira Alves et al: Ecdysone coordinates plastic growth with robust pattern in the developing wing | https://doi.org/10.26180/13393676 | figshare, 10.26180/13393676 |

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
