## [Editor Report]

This article is a carefully done assessment of the role of the moulting hormone ecdysone in coordinating growth and patterning of the wing imaginal disc in the final larval instar of *Drosophila* with nutritional input. Importantly, the authors find that growth is only partially dependent on the ecdysteroid titre, whereas the onset of bristle patterning is dependent on a threshold level that is different for different genes.

---

## [Decision Letter]

**Decision letter after peer review:**

Thank you for submitting the article "Ecdysone coordinates plastic growth with robust pattern in the developing wing" for consideration by *eLife*. Your article has been reviewed by 3 peer reviewers, including Lynn M. Riddiford as Reviewing Editor and Reviewer #1, and the evaluation has been overseen by K VijayRaghavan as the Senior Editor.

The reviewers have discussed their reviews with one another, and I have drafted this to help you prepare a revised submission.

Essential revisions:

Please see the Recommendations for the authors from each of the 3 reviewers below. Please try to answer all the comments in making your revision. For those comments with which you disagree, please write a rebuttal in your cover letter when you submit the revised manuscript.

*Reviewer #1 (Recommendations for the authors):*

Specific details that need clarification before publication:

1) It would help readers not familiar with advanced statistics to define the Gompertz function.

2) Although the authors use "ecdysone" to discuss the general function of the hormone, they should use 20E throughout the experimental section and discussion where they are using specific amounts of the hormone in the food to avoid possible reader confusion with the chemical "ecdysone", sometimes known as α-ecdysone.

3) Figure 6: The legend does not match the figure as to (A) and (B).

4) In Supplementary Figure 1, the data for + > grim appears to be nearly flat rather than showing a significantly positive correlation between ecdysone concentration in the food and the ecdysone concentration in the blood as the authors claim. You need to indicate that Supplementary Table 6 has the statistical parameters to support your claim.

5) Indicate in the figure legends for Figures 2, 3, and 7 and Supplementary Figures 1, 2, 4, and 5, how many animals were used for each point. Because of the overlap of many points, particularly the solid points so that there are different shades of gray or red, it is impossible to tell.

6) P24: The first full paragraph discussing the threshold response of patterning to ecdysone seems cumbersome and difficult to understand. Rewriting with some condensation would help.

7) Be consistent in formatting the references. In some the journal names are written out and in others they are abbreviated. Use this journal's format throughout.

*Reviewer #2 (Recommendations for the authors):*

– The data plotted in Figure 7 are somewhat noisy across 20E concentrations. The 12.5mg/mL and 25mg/mL concentrations are particularly ambiguous in their effects on disc size. This limits confidence in the conclusion that disc growth occurs through a graded response to 20E. While the models employed to fit the data result in a small p value, the shapes of the curves in Figure 7 and Figure 8 are not terribly dissimilar, despite different conclusions being drawn.

– A related comment, the authors state that 6.25mg/mL of 20E was insufficient to rescue Ac patterning, whereas 25mg/mL of 20E and above rescued Ac patterning, and from these data, they conclude that there is a threshold response to ecdysone. However, the authors also state that 12.5mg/mL showed intermediate levels of patterning. It would be helpful to explain how an intermediate level of patterning is distinct from a graded response.

– It is unclear why the authors focused on the first 20-hours after 3rd larval ecdysis for the experiments with defined concentrations of supplemental 20E, especially when they state elsewhere in the manuscript that wing disc growth seems to be independent of ecdysone at early 3rd instar stages. Would there be a more pronounced difference between 20E concentrations on wing disc growth at later stages of development?

– While the data suggest that wing disc growth exhibits a graded response to 20E, these data remain correlative. There is no demonstration that plasticity of wing disc growth is caused by a graded response to ecdysone. An alternative hypothesis is that growth plasticity is a consequence of ecdysone-independent signals. Interestingly, Figure 5A suggests that the PG secretes an inhibitory growth signal because starved larvae supplemented with 20E have smaller discs than PGX starved larvae supplemented with 20E. It is possible that ecdysone-independent nutritional signaling could determine plasticity of wing growth.

*Reviewer #3 (Recommendations for the authors):*

p9, Hypothesis 1 is Figure 1B not 1C.

In order to show the effectiveness of ablating prothoracic glands in reducing ecdysone level it would be good to extract and present from the experience represented in Supplementary figure 1, the ecdysone concentration in PGX and control larvae raised on normal diet.

---

## [Author Response]

Reviewer #1 (Recommendations for the authors):Specific details that need clarification before publication:1) It would help readers not familiar with advanced statistics to define the Gompertz function.

This is an excellent point. To try to clarify our description of the Gompertz function, we have added the following description to lines 210-212.

“Insect wing discs show damped exponential, or fast-then-slow, growth dynamics [63, 67]. These types of growth dynamics have frequently been modelled using a Gompertz function, which assumes that exponential growth rates slow down with time.”

2) Although the authors use "ecdysone" to discuss the general function of the hormone, they should use 20E throughout the experimental section and discussion where they are using specific amounts of the hormone in the food to avoid possible reader confusion with the chemical "ecdysone", sometimes known as α-ecdysone.

Thank you for raising this, we had indeed overlooked this detail. We have changed the figures and text to use 20E whenever we refer to experimental manipulations of ecdysteroids.

3) Figure 6: The legend does not match the figure as to (A) and (B).

Thanks very much, we have corrected this error.

4) In Supplementary Figure 1, the data for + > grim appears to be nearly flat rather than showing a significantly positive correlation between ecdysone concentration in the food and the ecdysone concentration in the blood as the authors claim. You need to indicate that Supplementary Table 6 has the statistical parameters to support your claim.

We have reanalysed this data, grouping the control genotypes as we have done throughout the manuscript for consistency. We further found that this data was better fit with second order polynomial function for ecdysone concentration, which allows the relationship between 20E in the hemolymph and diet to curve. We have also referred to the slope terms in Supplementary Table 8 to support our claim:

“There is a significant positive relationship between 20E concentration in the food and the concentration of ecdysteroids in the larval blood, as indicated by a significant 20E term (Supplementary Table 8).”

5) Indicate in the figure legends for Figures 2, 3, and 7 and Supplementary Figures 1, 2, 4, and 5, how many animals were used for each point. Because of the overlap of many points, particularly the solid points so that there are different shades of gray or red, it is impossible to tell.

The number of discs sampled/genotype have now been added to the legend of all plots.

6) P24: The first full paragraph discussing the threshold response of patterning to ecdysone seems cumbersome and difficult to understand. Rewriting with some condensation would help.

We have worked to streamline this paragraph, in the hopes that it improves its readability. Lines 918 – 931 now read:

“At first glance, the observation that patterning shows a threshold response to ecdysone may not be surprising. In any given cell, patterning is inherently regulated by threshold responses because the expression of the patterning gene product is either on or off in that cell. However, our patterning scheme considers the progression of patterning across the entire field of cells that make up the wing disc. Cells across the wing disc turn on Achaete and Senseless expression at different times, resulting in a continuous progression of pattern with time [42]. Furthermore, like growth, the progression of pattern can vary in rate depending on environmental and hormonal conditions [42]. Consequently, the progression of patterning could, in principle, also show a graded response to ecdysone levels. The observation that once ecdysone concentrations are above threshold, the rate of patterning for Achaete and Senseless is independent of ecdysone provides evidence that the rate of patterning across an entire organ can also show a threshold response: an assumption that, hitherto, has not been tested.”

7) Be consistent in formatting the references. In some the journal names are written out and in others they are abbreviated. Use this journal's format throughout.

Thanks very much for picking this up. We have corrected the references.

Reviewer #2 (Recommendations for the authors):– The data plotted in Figure 7 are somewhat noisy across 20E concentrations. The 12.5mg/mL and 25mg/mL concentrations are particularly ambiguous in their effects on disc size. This limits confidence in the conclusion that disc growth occurs through a graded response to 20E. While the models employed to fit the data result in a small p value, the shapes of the curves in Figure 7 and Figure 8 are not terribly dissimilar, despite different conclusions being drawn.

It’s not the shape of the curves that matter for determining graded versus threshold responses, but rather the distribution of the curves. To clarify this point, we have now plotted the linear growth rate for each 20E concentration, and fit this data with a Michaelis Menten model (Figure 7B). This model shows that as 20E concentrations increase, the linear growth rate also increases, although the rate of increase slows down at concentrations greater than 25 ng/ml. In contrast, Achaete and Senseless patterning rates of increase show switch -like responses that are best fit with a log-logistic models (please see the new figure panels Figure 7B, Figure 8B and D, Figure 9B and D, and accompanying text lines 522-807).

– A related comment, the authors state that 6.25mg/mL of 20E was insufficient to rescue Ac patterning, whereas 25mg/mL of 20E and above rescued Ac patterning, and from these data, they conclude that there is a threshold response to ecdysone. However, the authors also state that 12.5mg/mL showed intermediate levels of patterning. It would be helpful to explain how an intermediate level of patterning is distinct from a graded response.

We agree, the way this was originally phrased was confusing. We have addressed this by altering our analysis to focus on the key point highlighted above: does ecdysone after rates of growth and patterning as a graded or threshold response? By extracting the linear rates of increase from the growth and patterning dataset, we show that growth rates increase gradually with 20E concentration (Figure 7B). In contrast, Achaete and Senseless patterning rates are best fit with a threshold response function (Figure 8B and D, Figure 9B and D).

– It is unclear why the authors focused on the first 20-hours after 3rd larval ecdysis for the experiments with defined concentrations of supplemental 20E, especially when they state elsewhere in the manuscript that wing disc growth seems to be independent of ecdysone at early 3rd instar stages. Would there be a more pronounced difference between 20E concentrations on wing disc growth at later stages of development?

The reason for we focused on the first 20 hours of the third instar is that wing disc patterning becomes ecdysone independent around 10-15 hours. It is important to note that wing disc growth does depend on ecdysone in starved larvae in the first 20 hours, but the effects are not as strong in fed larvae.

– While the data suggest that wing disc growth exhibits a graded response to 20E, these data remain correlative. There is no demonstration that plasticity of wing disc growth is caused by a graded response to ecdysone. An alternative hypothesis is that growth plasticity is a consequence of ecdysone-independent signals. Interestingly, Figure 5A suggests that the PG secretes an inhibitory growth signal because starved larvae supplemented with 20E have smaller discs than PGX starved larvae supplemented with 20E. It is possible that ecdysone-independent nutritional signaling could determine plasticity of wing growth.

If we have understood the reviewer’s comments correctly, they state that the relationship between wing disc growth and 20E concentration is correlative. If we had measured ecdysteroid titres in the hemolymph and related these to wing disc growth rates, we would agree that the data is correlative. However, we experimentally manipulated 20E concentration and demonstrated that this had a graded response on wing disc growth rates in starved PGX larvae. Thus, we conclude that the effect of 20E on wing disc growth is causal.

To clarify this point, we have extracted the linear rate coefficients for wing disc growth, Achaete patterning, and Senseless patterning. We have then modelled the relationship between linear rates and 20E concentration for wing disc growth (new Figure 7B), Achaete patterning rate (new Figure 8B and D) and Senseless patterning rate (new Figure 9B and D). This is now discussed in Lines 522-807of the revised manuscript.

We agree that ecdysone-independent signals are also important for regulating plasticity of the wing disc, and make the point that nutrition is important for growth via ecdysone independent mechanisms throughout the manuscript. However, this manuscript focusses on how plasticity and robustness can be achieved through the action of the same developmental hormone, in this case ecdysone. The potential growth-inhibitory factor from the PG is very interesting, and is a topic of future research in the lab.

Reviewer #3 (Recommendations for the authors):p9, Hypothesis 1 is Figure 1B not 1C.

Yes, you are correct. Thank you for catching this.

In order to show the effectiveness of ablating prothoracic glands in reducing ecdysone level it would be good to extract and present from the experience represented in Supplementary figure 1, the ecdysone concentration in PGX and control larvae raised on normal diet.

Thank you very much for this recommendation, we have analysed the ecdysone titres in fed PGX and control larvae and found that PGX produce significantly lower titres than controls (Figure 2 Supplement 1).